# Multi-omics analyses and machine learning prediction of oviductal responses in the presence of gametes and embryos

Ryan M Finnerty[1], Daniel J Carulli[2], Akshata Hedge[3], Yanli Wang[3], Frimpong Boadu[3], Sarayut Winuthayanon[2], Jianlin Jack Cheng[3], Wipawee Winuthayanon[1,2]*

[1]Department of OB/GYN & Women's Health, School of Medicine, University of Missouri-Columbia, Columbia, United States; [2]Division of Animal Sciences, College of Agriculture, Food and Natural Resources, University of Missouri-Columbia, Columbia, United States; [3]Department of Electrical Engineering and Computer Science, College of Engineering, University of Missouri, Columbia, United States

## eLife Assessment

This **important** study reports the transcriptomic and proteomic landscapes of the oviducts at four different preimplantation stages during natural fertilization, pseudopregnancy, and superovulation. The supporting data are **convincing**. This work will be of interest to reproductive biologists and clinicians practicing reproductive medicine.

*For correspondence:
w.winuthayanon@health.
missouri.edu

Competing interest: The authors declare that no competing interests exist.

**Abstract** The oviduct is the site of fertilization and preimplantation embryo development in mammals. Evidence suggests that gametes alter oviductal gene expression. To delineate the adaptive interactions between the oviduct and gamete/embryo, we performed a multi-omics characterization of oviductal tissues utilizing bulk RNA-sequencing (RNA-seq), single-cell RNA-sequencing (scRNA-seq), and proteomics collected from distal and proximal at various stages after mating in mice. We observed robust region-specific transcriptional signatures. Specifically, the presence of sperm induces genes involved in pro-inflammatory responses in the proximal region at 0.5 days post-coitus (dpc). Genes involved in inflammatory responses were produced specifically by secretory epithelial cells in the oviduct. At 1.5 and 2.5 dpc, genes involved in pyruvate and glycolysis were enriched in the proximal region, potentially providing metabolic support for developing embryos. Abundant proteins in the oviductal fluid were differentially observed between naturally fertilized and superovulated samples. RNA-seq data were used to identify transcription factors predicted to influence protein abundance in the proteomic data via a novel machine learning model based on transformers of integrating transcriptomics and proteomics data. The transformers identified influential transcription factors and correlated predictive protein expressions in alignment with the in vivo-derived data. Lastly, we found some differences between inflammatory responses in sperm-exposed mouse oviducts compared to hydrosalpinx Fallopian tubes from patients. In conclusion, our multi-omics characterization and subsequent in vivo confirmation of proteins/RNAs indicate that the oviduct is adaptive and responsive to the presence of sperm and embryos in a spatiotemporal manner.

## Introduction

Optimal physiological conditions in the oviduct (fallopian tube in humans) provide an adaptive microenvironment for several reproductive processes ranging from sperm capacitation and transport to fertilization and embryonic development (*Li and Winuthayanon, 2017*). The oviduct comprises four

main regions: infundibulum (responsible for oocyte pick-up), ampulla (site of fertilization), isthmus (sperm capacitation/transport and preimplantation embryonic development), and the uterotubal junction (UTJ; responsible for filtering sperm and embryo transit to the uterus). Several studies demonstrated that distal (infundibulum and ampulla: IA) and proximal (isthmus and UTJ: IU) regions of the oviduct have distinct transcriptional profiles (*Roberson et al., 2021*; *McGlade et al., 2021*; *Ford et al., 2021*; *Harwalkar et al., 2021*). However, it is unclear how the presence of the sperm and embryo(s) modulates the oviductal responses. The presence of gametes and embryos has been shown to alter gene expression in secretory and ciliated cells of the oviduct during the preimplantation period (*Maillo et al., 2015*; *Almiñana et al., 2012*; *Lee et al., 2002*; *Fazeli et al., 2004*). Additionally, it was reported that the endometrium responded differently to in vivo-derived embryos compared to embryos derived from in vitro fertilization or somatic cell nuclear transfer in large animal models (*Mansouri-Attia et al., 2009*; *Bauersachs et al., 2009*), suggesting a maternal response to the presence of different types of embryos. Indeed, variations in the relative abundance of sets of genes involved in compaction and cavitation, desmosomal glycoproteins, metabolism, mRNA processing, stress, trophoblastic function, and growth and development have been observed in in vitro-produced embryos compared to their in vivo counterparts (*Kropp and Khatib, 2015*; *Smith et al., 2009*; *Tremoleda et al., 2003*; *Niemann and Wrenzycki, 2000*). Lastly, a growing consensus in several species indicates that epigenetic events in preimplantation embryos contribute to altered developmental potential both early and later in life (*Fleming et al., 2004*).

Reciprocal embryo–oviduct interactions stem largely from investigating oviductal transport of fertilized/unfertilized embryos/oocytes in livestock (*Maillo et al., 2015*; *Almiñana et al., 2012*; *Betteridge and Mitchell, 1974*; *Freeman et al., 1992*; *Lazzari et al., 2010*; *Leemans et al., 2016*; *Maillo et al., 2016a*; *Maillo et al., 2016b*; *Kues et al., 2008*; *Rodríguez-Alonso et al., 2020*; *Rodríguez-Alonso et al., 2019*; *Smits et al., 2016*; *Smits et al., 2017*; *Weber et al., 1991a*; *Weber et al., 1991b*; *Marey et al., 2016*) and rodents (*Roberson et al., 2021*; *Lee et al., 2002*; *Ortiz et al., 1986*; *Ortiz et al., 1989*). In humans, an embryo-derived platelet-activating factor has been implicated in the control of embryo transport to the uterus (*Velasquez et al., 2001*). It has been suggested that fertilized embryos produce prostaglandin E2 that facilitates transport to the uterus in mares (*Kues et al., 2008*; *Rodríguez-Alonso et al., 2019*), whereas non-fertilized eggs remain in the oviduct (*Betteridge and Mitchell, 1974*). In hamsters, fertilized embryos are transported more expeditiously to the uterus compared to unfertilized eggs (*Ortiz et al., 1986*). In rats, transferred advanced-stage embryos (4-cell vs. 1-cell) arrive in the uterus prematurely (*Ortiz et al., 1989*). In pigs (*Almiñana et al., 2012*) and cows (*Maillo et al., 2015*), pro-inflammatory responses in the oviduct are downregulated by the presence of embryos, suggesting that the embryo may facilitate maternal embryo tolerance during its passage through the oviduct. However, alterations in the oviductal transcriptome are difficult to detect in mono-ovulatory species (*Maillo et al., 2016b*; *Rodríguez-Alonso et al., 2019*) indicating that the effect of the embryo in the oviduct is localized.

As for the sperm, observations suggest a filtering process as sperm migrates from the uterus, through the UTJ, into the oviduct (*Krzanowska, 1974*; *Pérez-Cerezales et al., 2018*). After entering through the UTJ, sperm interact with ciliated cells in the isthmus to form a reservoir, undergo capacitation and are subsequently released to initiate the acrosomal reaction prior to reaching the ampulla (*Pérez-Cerezales et al., 2018*; *Arthur and Ley, 2013*; *Abe, 1996*). However, sperm are allogenic to the female reproductive tract, as sperm have been observed to induce pro- and anti-inflammatory responses in the oviduct (*Georgiou et al., 2007*; *Yousef et al., 2016*). Additionally, phagocytic bodies in the luminal fluid at the isthmus region can engulf sperm for degradation in mice (*Chakraborty and Nelson, 1975*). In addition to sperm selection, the oviduct seemingly provides beneficial chemical and mechanical mechanisms through rheotaxis, thermotaxis, and chemotaxis that assist sperm in transportation and fertilization (*Perez-Cerezales et al., 2015*; *Miki and Clapham, 2013*; *Oliveira et al., 1999*). These observations suggest that the oviduct provides a malleable environment that is plastic and adaptable to select and facilitate the fittest sperm for fertilization. Therefore, our study also intends to provide a better understanding of the oviductal environment before, during, and after the presence of sperm in different regions of the oviduct.

In recent years, the field of reproductive biology has increasingly leveraged artificial intelligence (AI) and machine learning technologies to delve deeper into the intricate mechanisms governing fertilization and embryonic growth. AI predictive models, such as powerful transformer models, have shown

remarkable capabilities in analyzing large-scale biological data, encompassing multi-omics data, to unveil patterns and forecast outcomes with elevated precision (*Wang et al., 2024*). One of the critical attributes of transformer models is the attention mechanism, which empowers the model to focus on pertinent essential segments of the input data that are critical for predicted outcomes (*Vaswani et al., 2017*). This functionality proves advantageous in the domain of reproductive biology, wherein complex interplays among genes, proteins, and other biomolecules dictating fertility outcomes may be revealed by the attention mechanism. The objective of this investigation is to amalgamate a multi-omics strategy with a transformer-based AI predictive model to elucidate the adaptive characteristics of the oviduct during natural fertilization.

Based on this premise, our study aims to elucidate the adaptive nature of the oviduct using a multi-omics approach during natural fertilization and preimplantation embryo development in a mouse model. We dissected oviducts from naturally fertilized mice at 0.5, 1.5, 2.5, and 3.5 days post-coitus, pseudopregnancy, and superovulation (dpc, dpp, and SO, respectively). Gene expression profiles were analyzed from two different regions of the oviduct (IA and IU) using bulk RNA and single-cell RNA (scRNA) sequencing (seq) analyses, generating a spatiotemporal depiction of gene expression in the oviduct. Observations of RNA expression profiles from bulk RNA-sequencing (RNA-seq) findings were reinforced by single-cell RNA-sequencing (scRNA-seq) and LC–MS/MS proteomics analysis. Lastly, we integrated bulk RNA-seq and proteomics datasets to develop the initial stages of a machine learning predictive model, which can identify influential transcription factors and correlate predictive protein expressions based on in vivo-derived data. Overall, we observed a robust transition of transcripts in the oviduct after sperm exposure at 0.5 dpc to other timepoints during preimplantation in both IA and IU regions. One of our key observations, was an elevated pro-inflammatory transcriptional and proteomic profile at 0.5 dpc, likely due to the presence of sperm preceding an anti-inflammatory condition 24 hr later, correlating with the spatial presence of the embryo in the IU region at 1.5 dpc. Furthermore, this study paves the way for formulating a pioneering integrative AI model methodology tailored to integrate transcriptomics and proteomics data.

## Results

### Bulk RNA-seq analysis reveals a dynamic transcriptional profile during pregnancy that exhibits a distinct signature from pseudopregnancy

To ensure the presence and location of embryos/eggs in the oviduct in our model, we sampled the oviduct at different timepoints and evaluated the location of the embryos/ovulated eggs using hematoxylin and eosin (H&E) staining. Fertilized and unfertilized eggs with surrounding cumulus cells were in the ampulla at 0.5 dpc/dpp, respectively (*Figure 1A*). Two-cell embryos and unfertilized eggs were clustered in the isthmus at 1.5 dpc/dpp. At 2.5 dpp/dpc, unfertilized eggs and embryos at the 8-cell to the morula stage were halted in a single-file formation at the UTJ region. At 3.5 dpc, the UTJ region was devoid of embryos/oocytes as all embryos/oocytes were transported to the uterus and, therefore, not included in the figure.

To determine whether the transcriptional profiles of each oviductal region are unique at fertilization and different developmental stages during preimplantation development, bulk RNA-seq analysis was performed at 0.5, 1.5, 2.5, and 3.5 dpc. Additionally, we aim to address whether changes in transcriptional signatures in the oviduct are governed by hormonal fluctuations or the presence of sperm/embryos/eggs. Therefore, oviducts from females at corresponding days post-mating with vasectomized males at (0.5, 1.5, 2.5, and 3.5 dpp) were used for comparisons. Principal component analysis (PCA) plots were generated using the top 2500 differentially expressed genes (DEGs, *Figure 1—figure supplement 1A, B*). Broad observations of region-specific transcriptome uniqueness exhibited segregation of all IA and IU biological replicates to opposite ends of the center axis on the PC1, reinforcing previous findings (*Harwalkar et al., 2021*) that IA and IU regions behave differently with respect to transcriptional activity. Surprisingly, with respect to both the IA and IU regions, overall transcripts at 0.5 dpc (*Figure 1B*) were segregated to the topmost axis along the PC2 plane, while 1.5–3.5 dpc biological replicates were segregated to the bottommost axis along the PC2 plane.

Expression signatures of the top 2500 DEGs in the IA region during pseudopregnancy were similar to those during pregnancy, as indicated by a heatmap generated using unsupervised hierarchical clustering (*Figure 1—figure supplement 1C, D*). However, there were exceptions at 0.5 (*Figure 1—figure*

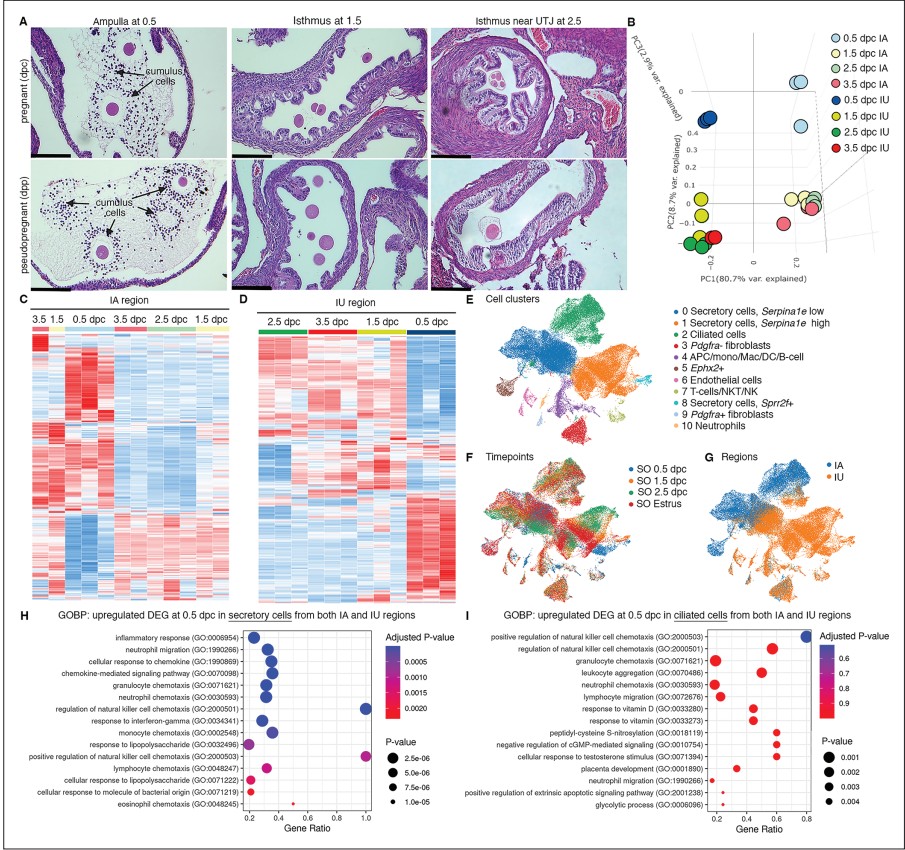

**Figure 1.** Transcriptomic analyses of the oviduct at different stages of early pregnancy. (**A**) Histological analysis of different oviductal regions (ampulla, isthmus, and near the uterotubal junction (UTJ)) in mice at different stages of pregnancy (0.5, 1.5, and 2.5 dpc) and pseudopregnancy (0.5, 1.5, and 2.5 dpp) using hematoxylin and eosin (H&E) staining (scale bars = 132 μm, *n* = 3 mice/timepoint/region). Arrows indicate cumulus cells surrounding the eggs/fertilized eggs called cumulus–oocyte complexes. (**B**) Principal component analysis (PCA) of top 2500 differentially expressed genes (DEGs) identified from bulk RNA-sequencing (RNA-seq) of the infundibulum + ampulla (IA) and isthmus + UTJ (IU) regions of the oviduct collected at 0.5, 1.5, 2.5, and 3.5 dpc. Heatmap plots of unsupervised hierarchical clustering of top 2500 DEGs identified from bulk RNA-seq in the oviduct during pregnancy (0.5, 1.5, 2.5, and 3.5 dpc) of (**C**) IA and (**D**) IU regions. (**E, F**) Single-cell RNA-sequencing (scRNA-seq) analysis of the oviduct from superovulated (SO) estrus, SO 0.5 dpc, SO 1.5 dpc, and SO 2.5 dpc. Uniform manifold approximation and projection (UMAP) of (**E**) cell clusters identified from the oviduct (**F**) at different regions (IA and IU) and (**G**) at different timepoints (*n* = 3–4 mice/timepoint/region). Gene ontology biological processes (GOBPs) dot plots of scRNA-seq analysis when compared between upregulated DEGs from (**H**) secretory epithelial cells and (**I**) ciliated epithelial cells at SO 0.5 dpc compared to SO estrus from both IA and IU regions.

The online version of this article includes the following figure supplement(s) for figure 1:

**Figure supplement 1.** Transcriptomic profiles of mouse oviducts during pregnancy and pseudopregnancy.

**Figure supplement 2.** Dot Plots of biological processes from gene ontology (GOBP) analysis from differentially expressed genes (DEGs) from bulk RNA-sequencing (RNA-seq) analysis at different timepoints.

**Figure supplement 3.** Analyses of differentially expressed genes that were enriched at different timepoints.

---

*supplement 1C*, blue box) and 1.5 (*Figure 1—figure supplement 1C*, black box) dpc/dpp. Unlike the IA region, DEGs in the IU region were more dynamic, as indicated by the presence of unique sets of genes at 0.5, 1.5, 2.5, and 3.5 dpc timepoints between pregnancy versus pseudopregnancy (*Figure 1—figure supplement 1D*, blue, black, and red boxes). These findings indicate that oviduct transcripts from pregnant mice also possessed distinct signatures from pseudopregnant samples. Overall, data suggest that the transcriptional profile in the oviduct at all stages during the preimplantation period in the IU region is more dynamic compared to the IA region.

## Cellular responses to inflammation are enriched at the proximal (IU) and distal (IA) regions in response to the sperm

Oviductal transcription signatures were more unique at 0.5 dpc when compared to those at 1.5–3.5 dpc (i.e., 0.5 dpc vs. rest) in both IA and IU regions (*Figure 1C, D*). To determine the biological process of genes that were differentially expressed at 0.5 dpc compared to 1.5–3.5 dpc in both IA and IU regions, an initial analysis (0.5 dpc vs. rest) was chosen to isolate what distinct processes may be occurring during the transition from 0.5 dpc. Unique DEGs upregulated at 0.5 dpc in the IA region were enriched for the following biological processes (BPs): extracellular matrix (ECM) organization, extracellular structure organization, collagen fibril organization, and $Ca^{2+}$ ion homeostasis, among others (*Figure 1—figure supplement 2A*). Most interestingly, we found upregulated DEGs enriched for BPs at 0.5 dpc in the IU region included cellular response to cytokine stimulus, neutrophil migration, response to interferon-gamma, response to lipopolysaccharide, and neutrophil chemotaxis (*Figure 1—figure supplement 2B*). Moreover, there are multiple BPs involved in the glucose catabolic process to pyruvate, in addition to other pyruvate metabolic processes that were uniquely upregulated at 0.5 dpc in the IU region (*Figure 1—figure supplement 2B*). Next, we evaluated the IA region at 0.5 dpc compared to 1.5 dpc (presence of sperm vs. 24 hr post-sperm exposure in the presence of embryos). We observed significant BPs that were enriched for downregulated DEGs at the IA region at 1.5 dpc compared to 0.5 dpc (*Figure 1—figure supplement 2C, D*). These processes included response to interferon-gamma, neutrophil chemotaxis, cytokine-mediated signaling pathway, and neutrophil migration. BPs common to the IA region at 0.5 dpc also included ECM organization, extracellular structure organization, and collagen fibril organization.

DEGs were more dynamic in the IU region during preimplantation embryo development compared to the IA region. At 0.5 dpc, the sperm are present, creating a sperm reservoir in the isthmus (*Demott and Suarez, 1992*). When comparing 0.5 dpc to 0.5 dpp in the IU region (presence or absence of sperm, respectively), gene ontology biological processes (GOBPs) analysis revealed significant enrichment of multiple pro-inflammatory BPs, including inflammatory response, neutrophil migration, neutrophil chemotaxis, regulation of phagocytosis, positive regulation of acute inflammatory response, and response to lipopolysaccharide when sperm were present in the IU (*Figure 1—figure supplement 2E, F*). Therefore, it is likely that, at 0.5 dpc, the isthmus region of the oviduct was heavily regulated for an inflammatory response in the presence of sperm while simultaneously preparing for the metabolic switch of the embryos from pyruvate to glucose metabolism.

Next KEGG analysis was used to determine molecular players; we found that genes in the tumor necrosis factor (TNF) signaling pathway were mostly upregulated at 0.5 dpc compared to 0.5 dpp at the IU region (*Figure 1—figure supplement 3A*). Subsequently, an analysis comparing 0.5–1.5 dpc in the IU region (presence of sperm vs. presence of embryos) demonstrated the most striking results. We found that most upregulated genes at 0.5 dpc were now downregulated at 1.5 dpc (*Figure 1—figure supplement 3B*). Many of these genes are involved in the cellular response to cytokine stimulus, response to interferon-gamma, response to lipopolysaccharide, and neutrophil chemotaxis. These data strongly suggest that the oviduct may suppress the response to inflammation in the isthmus once the sperm is cleared to become conducive for the embryo's survival at 1.5 dpc.

## scRNA-seq reveals that secretory epithelial cells contribute to the pro- and anti-inflammatory responses in the oviduct

To determine the key cell types responsible for the oviductal response to sperm/embryos, scRNA-seq analyses were leveraged. As we did not observe significant transcriptional changes from bulk RNA-seq at 3.5 dpc and embryos were not present in the oviduct at 3.5 dpc, we opted not to assess a 3.5 dpc timepoint in our scRNA-seq analysis. In this experiment, SO using exogenous gonadotropins was used due to technical limitations of sample collection for single-cell processing. Non-mated SO estrus samples were used as controls. First, we confirmed that all cell types previously reported (*McGlade et al., 2021*) were present in the oviduct (*Figure 1E, F*). There was minimum overlap between cells isolated from IA or IU regions (*Figure 1G*). In addition, all cell types were present at all timepoints except for an *Ephx2+* cluster (only present at SO 0.5 dpc and SO estrus) and a neutrophil cluster (*Ly6g+*, only present at SO 0.5 dpc).

Next, we investigated whether our findings from bulk RNA-seq data would be recapitulated in the scRNA-seq dataset. Here, we exclusively focused on the IU region as it was the most dynamically

regulated region during early pregnancy. Based on GOBP analysis from bulk RNA-seq findings, we further assessed several genes that were upregulated at 0.5 and 1.5 dpc corresponding to GOBP terms 'inactivation of mitogen-activated protein kinase (MAPK) activity' and 'MAP kinase phosphatase activity'. Genes associated with these pathways were mostly upregulated at SO 0.5 and SO 1.5 dpc (*Figure 1—figure supplement 3C*, green and teal bars) and downregulated in SO estrus and SO 2.5 dpc in the IU regions (*Figure 1—figure supplement 3D*, red and purple bars). These genes include dual-specificity phosphatase family (*Dusp1*, *Dusp5*, *Dusp6*, *Dusp10*), *Fos*, interleukin 1b (*Il1b*), IL1 receptor 2 (*Il1rb*), and others. As DUSP proteins are crucial for controlling inflammation and antimicrobial immune responses (*Arthur and Ley, 2013*), we performed qPCR analysis to confirm both our bulk RNA- and scRNA-seq data with respect to MAPK signaling pathways. We found that *Dusp5* was expressed at a significantly higher level at 0.5 dpc compared to 0.5 dpp while *Mapk14* (*p38a*) was significantly upregulated at 1.5 dpc compared to 0.5 dpc (*Figure 1—figure supplement 3E, F*). We also further assessed several genes from the GOBP term 'neutrophil-mediated immunity' to explain the appearance at SO 0.5 dpc and subsequent disappearance of the neutrophil cluster at SO 1.5 dpc, respectively (*Figure 1E, F*). Interestingly, these genes were found to be downregulated in both the IU and IA regions at the 1.5 and 2.5 dpc timepoints (*Figure 1—figure supplement 3G, H*, teal and purple bars).

To identify which cell population is contributing to the observed pro- and anti-inflammatory response in both IA and IU regions at SO 0.5 dpc. We evaluated DEGs in each cell population and performed GOBP analysis. Upregulated DEGs from secretory epithelial cells (both clusters 0 and 1; *Figure 1E–I*) from both IA and IU regions at 0.5 dpc were enriched for BPs involved in inflammatory response, neutrophil migration, cellular response to chemokine, chemokine-mediated signaling pathways, and several others chemokine signaling pathways (*Figure 1H*). In contrast, when evaluated for upregulated DEGs in ciliated epithelial cells, similarly enriched biological processes were present, albeit in a less significant manner (*Figure 1I*). Therefore, it suggests that secretory cells are the key modulators responsible for the regulation of pro- and anti-inflammatory responses during pregnancy establishment.

## Oviductal luminal proteomics are dynamic at different preimplantation stages and SO exacerbates the transcriptional profile at each timepoint

To validate our transcriptomics data at a translational level, LC–MS/MS proteomic analysis was performed on secreted proteins in the oviductal luminal fluid at estrus, 0.5, 1.5, and 2.5 dpc with or without SO. As we also aim to address whether changes in proteomic profiles in the oviduct are governed by hormonal fluctuations, the SO was performed using exogenous gonadotropins. Therefore, the comparison was assessed in the following groups: estrus, 0.5 dpc, 1.5 dpc, 2.5 dpc, SO estrus, SO 0.5 dpc, SO 1.5 dpc, and SO 2.5 dpc. In agreement with the transcriptomic data, secreted proteins from 0.5 dpc and SO 0.5 dpc were segregated from all other timepoints (*Figure 2A–C*). Another difference was observed between 1.5 dpc and SO 1.5 dpc, at which 1.5 dpc proteomic dynamics correlated more with estrus and SO estrus biological replicates, while SO 1.5 dpc correlated more with 2.5 dpc and SO 2.5 dpc.

Analysis comparing naturally fertilized (dpc) pregnant samples (without SO) yielded 242 differentially abundant proteins between estrus and 0.5 dpc, 185 between 0.5 and 1.5 dpc, and 344 between 1.5 and 2.5 dpc (*Figure 2D*). Next, we elucidated whether SO treatment impacts protein secretion in the oviduct. There were 298, 354, and 163 differentially abundant proteins when compared between SO estrus versus SO 0.5 dpc, SO 0.5 dpc versus SO 1.5 dpc, and SO 1.5 dpc versus SO 2.5 dpc, respectively (*Figure 2E*). In addition, protein samples from naturally fertilized and SO samples were evaluated. There were 112 differentially abundant proteins between estrus and SO estrus, 111 between 0.5 dpc and SO 0.5 dpc, 371 between 1.5 dpc and SO 1.5 dpc, and 274 between 2.5 dpc and SO 2.5 dpc (*Figure 2F*). These results indicate that luminal proteomics from the oviduct are dynamic during preimplantation development and SO stimulates the production and secretion of more abundant and unique proteins compared to the natural setting.

Next, we explored differentially abundant proteins commonly shared between estrus versus 0.5 dpc and 0.5 dpc versus 1.5 dpc. We found that a subset of shared 100 proteins were enriched for multiple pro-inflammatory Reactomes including neutrophil degranulation, innate immune system, and innate immune system (*Figure 2G, I*). In addition, when evaluated a subset of shared 105 protein samples

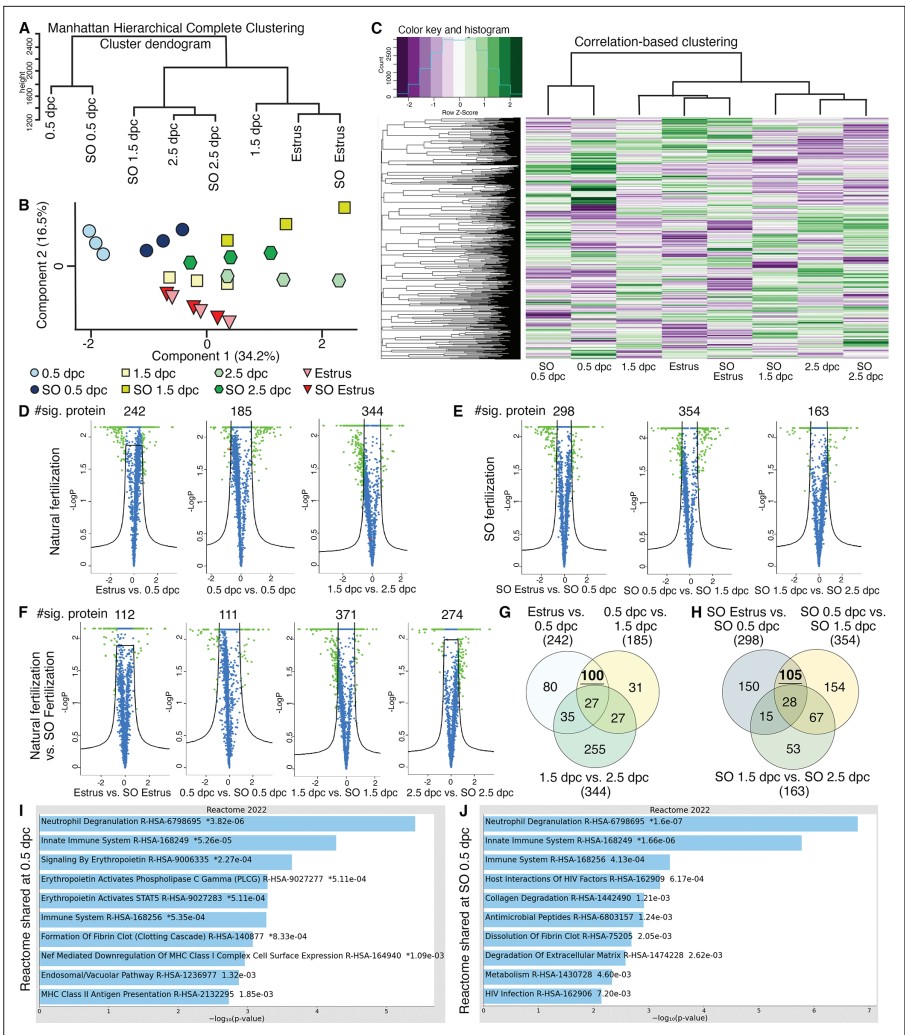

**Figure 2.** Analyses of protein abundance in the oviduct luminal fluid at different stages of pregnancy compared to Estrus. (**A**) Manhattan hierarchical complete clustering dendrogram of natural (estrus, 0.5 dpc, 1.5 dpc, and 2.5 dpc) and superovulated (SO estrus, SO 0.5 dpc, SO 1.5 dpc, and SO 2.5 dpc) datasets (*n* = pooled of 3 biological samples/timepoint). (**B**) Principal component analysis (PCA) plot of all datasets generated utilizing Perseus software after integration of the Gaussian transformation. (**C**) Correlation-based hierarchal clustering of all protein abundance. Volcano plots of significantly different protein abundances when compared between (**D**) natural fertilization, (**E**) SO fertilization, and (**F**) natural fertilization versus SO fertilization. Numbers of significant proteins were listed above the volcano plots. Gaussian transformed Perseus two-tailed *t*-tests of differentially abundant proteins in oviductal fluid at different stages during (**G**) natural fertilization and (**H**) SO fertilization. Differentially abundant proteins shared between estrus and 0.5 dpc (100) or SO estrus and SO 0.5 dpc (105) were underlined. Enricher Reactome pathway analysis of differentially abundant proteins shared at (**I**) 0.5 and (**H**) SO 0.5 dpc.

The online version of this article includes the following figure supplement(s) for figure 2:

**Figure supplement 1.** Enricher Reactome pathway analysis of differentially expressed proteins in the luminal fluid at (**A**) 1.5 and (**B**) 2.5 dpc.

---

with SO at the same timepoints, similar if not identical Reactomes occurred, with lower p-values (*Figure 2H, J*), indicating greater pathway enrichment in SO treatments. Lastly, differential protein abundance at 1.5 and 2.5 dpc indicated the enrichment for Ras Homolog (RHO) GTPase signaling pathway and changes in epithelial remodeling (keratinization) (*Figure 2—figure supplement 1A, B*), respectively. Therefore, the pro-inflammatory Reactome profile appeared to have completely subsided at 2.5 dpc. These results reinforce our bulk and scRNA-seq observations of a pro-inflammatory condition occurring at 0.5 dpc. Moreover, SO conditions appear to exacerbate both expression abundance

and the expression of additional unique proteins with respect to pro-inflammation when compared to naturally fertilized replicates.

## In vivo confirmation of identified multi-omics pro-inflammatory condition in the oviduct at 0.5 dpc

To validate the findings from multi-omics studies, we used RNAScope in situ hybridization staining of *Tlr2* (epithelium, stroma, and myosalpinx), *Ly6g* (leukocytes), and *Ptprc* (common immune cell marker). We found a significant induction of *Tlr2* at 0.5 dpc compared to 0.5 dpp at the isthmus and UTJ regions (*Figure 3A, B*). *Ptprc+* and *Ly6g+* signals aggregated with greater intensities in the mesosalpinx, stromal layer, and blood vessels in the oviduct. Additionally, no positive *Ly6g+* cell expression was found in the luminal space of the oviduct, but rather restricted to stromal and epithelial cell linings along with blood vessels.

NFκB immunofluorescent staining was performed to evaluate the degree of inflammatory activation. The presence of NFκB appeared to be largely concentrated in the cytoplasm of all epithelial cells at all timepoints in the isthmus. The relative fluorescent signal was significantly greater at 0.5 dpc compared to 1.5 dpc or 0.5 dpp (*Figure 3C, D*). As p38 is the key mediator of the inflammatory response (*Yang et al., 2014*), we found that p38 and phosphorylated (p)-p38 proteins were expressed at all timepoints between 0.5 and 1.5 dpc and dpp (*Figure 3E, F*). Specifically, p-p38:total p38 ratio was significantly increased at 0.5 dpp compared to 0.5 dpc, suggesting an overall inflammatory response induced by mating regardless of the sperm exposure. In addition, the presence of pro-inflammatory cytokine, IL1β was evaluated. However, there was no difference in IL1β levels between timepoints (*Figure 3G*). To summarize, these data suggest that an innate immune response occurs at 0.5 dpc in the isthmus and UTJ regions and that some of these responses were induced by the presence of seminal plasma regardless of the sperm.

## Integrating transcriptomics and proteomics data and identifying influential transcription factors in the oviduct via a predictive transformer model

Our machine learning method based on a transformer encoder model is rigorously evaluated against gene and protein expression data from 2.5 dpc of naturally fertilized samples, which was not used by the model during its training. The integrative transformer model was effective in predicting the protein abundance levels from bulk RNA-seq expression data with high accuracy. The attention matrix for all genes against all proteins is extracted from the transformer model, which represents each gene's potential influence level on the proteins (*Figure 4A*). To focus on analyzing DEGs and proteins rather than all the genes and all proteins, differential gene expression and protein abundance expression between bulk RNA-seq and proteomic datasets at 0.5, 1.5, and 2.5 dpc were compared to Estrus and proteomics Estrus, respectively, followed by extraction of common significantly differentiated protein-coding genes or proteins (*Figure 4B*). The differential gene expression is performed using DESeq2 (*Liu et al., 2021*) and the differential protein abundance analysis using Protrank (*Medo et al., 2019*). The top 25 'influential' transcripts (ITs) with the highest attention scores from all the transcription factors present in bulk RNA-seq data were extracted for every potentially influenced protein (IP) in the empirical proteomics datasets (*Supplementary file 1*, *Supplementary file 2*, *Supplementary file 3*, and *Supplementary file 4*).

The identified IT and IP lists were subsequently analyzed with Enrichr Reactome (2022) and GO Biological Process (2023) tools. A combination of both IT and IP lists generated function ontologies that match in vivo empirical observations. At 0.5 dpc, ITs predicted to influence protein abundance included, but are not limited to, *Clu*, *Anxa2*, *Nod2*, *Hspa8*, *Il17c*, *Il36b*, and *Il1b*, among many others (*Supplementary file 2*). Among the top 25 ITs identified in high abundance at 1.5 dpc included *Cep126*, *Cfap126*, *Cfap54*, *Cfap65*, *Ift88*, *Ccdc40*, *Crocc2*, and *Clu* (*Supplementary file 3*). Lastly, ITs abundant at 2.5 dpc included, but were not limited to, *Mapk15*, *Hsph1*, *Drc7*, *Togaram2*, *Tspan15*, *Igfbp2*, *Rnf112*, and *Traf3ip3* (*Supplementary file 4*). Taken together, we have developed a predictive transformer model that has recapitulated a similar progressive observation as our in vivo empirically biological multi-omics model. Moreover, the predictive model suggests that ITs and IPs present at 0.5 dpc indicate a pro-inflammatory condition, followed by a shift to ciliogenesis and cellular stress maintenance at 1.5 dpc, subsequently initiating cellular homeostasis at 2.5 dpc. In addition, this predictive

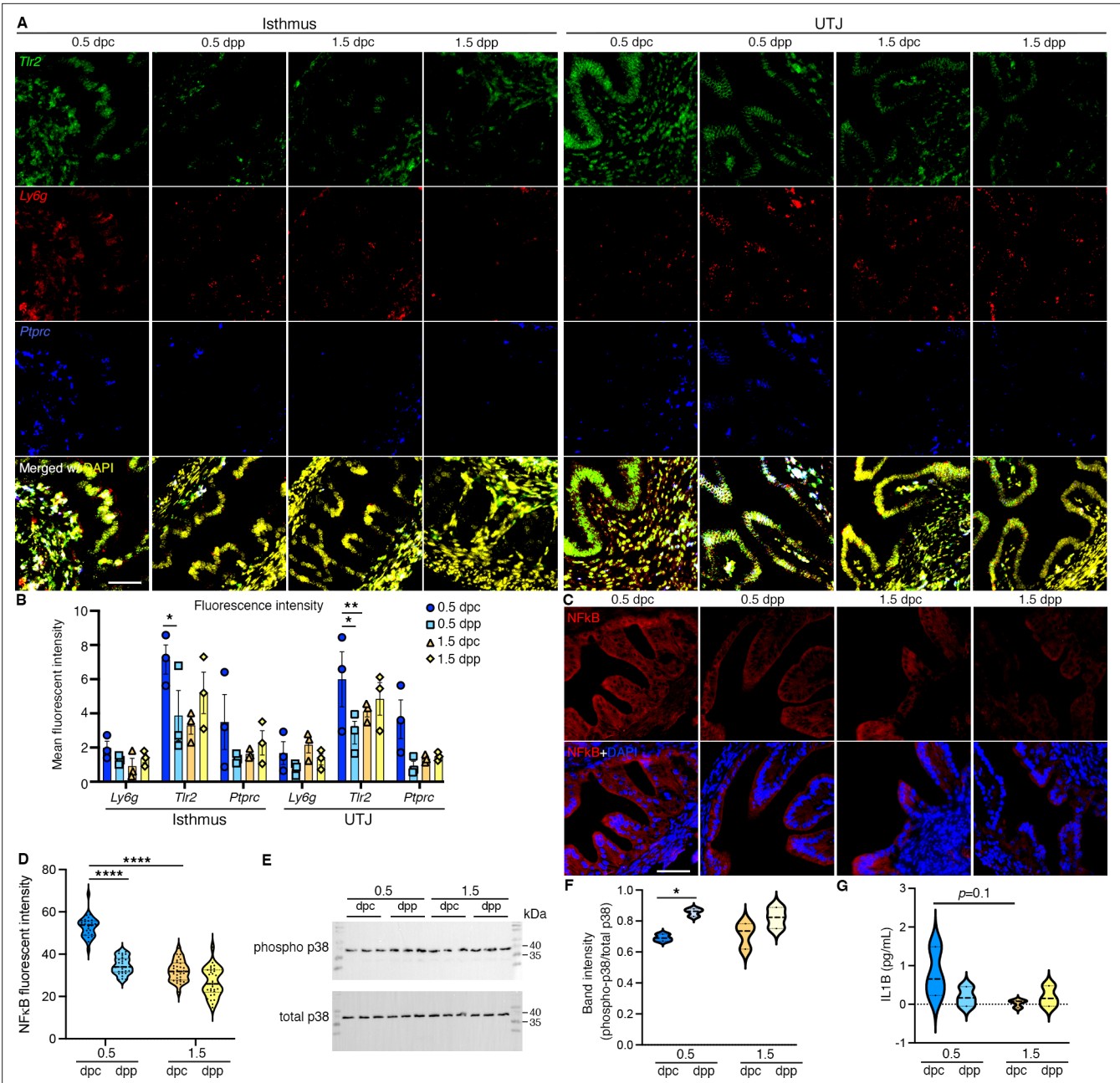

**Figure 3.** In vivo validation of RNA and proteins identified from bulk RNA-sequencing (RNA-seq) and single-cell RNA-sequencing (scRNA-seq) analysis.
(**A, B**) Expression of *Tlr2*, *Ly6g*, and *Ptprc* in the isthmus and uterotubal junction (UTJ) regions at 0.5 dpc, 1.5 dpc, 0.5 dpp, and 1.5 dpp. Scale bar = 50 μm for all images in the panel. (**B**) Quantification of fluorescent signal from images in (**A**) using FIJI software. Graph represent mean ± SEM, $n$ = 3 mice/timepoint/region. *,**$p < 0.05$ or 0.01 respecitvely compared to 0.5 dpc, unpaired $t$-test. (**C**) Immunofluorescent staining of NF$\kappa$B in the isthmus regions of the oviducts at 0.5 dpc, 1.5 dpc, 0.5 dpp, and 1.5 dpp. Scale bar = 50 μm for all images in the panel. (**D**) Quantification of fluorescent signal from images in (**C**) using FIJI software. Violin plots represent all measurements, $n$ = 3 mice/timepoint/region, ****$p < 0.001$ compared to 0.5 dpc, unpaired $t$-test. (**E**) Immunoblotting of phosphorylated p38 and total p38 in the whole oviduct collected at 0.5 dpc, 0.5 dpp, 1.5 dpc, and 1.5 dpp. (**F**) Violin plots of the quantification of band intensity represented as phosphor-p38/total p38 ratio ($n$ = 3 mice/timepoint/region). *$p < 0.05$ compared to 0.5 dpc, unpaired $t$-test. (**G**) IL1β ELISA of protein from the whole oviduct at 0.5 dpc, 1.5 dpc, 0.5 dpp, and 1.5 dpp ($n$ = 3 mice/timepoint/region).

The online version of this article includes the following source data for figure 3:

**Source data 1.** Original files for western blot analysis displayed in *Figure 3E*.

**Source data 2.** PDF file containing original western blots for *Figure 3E*, indicating the relevant bands and timepionts.

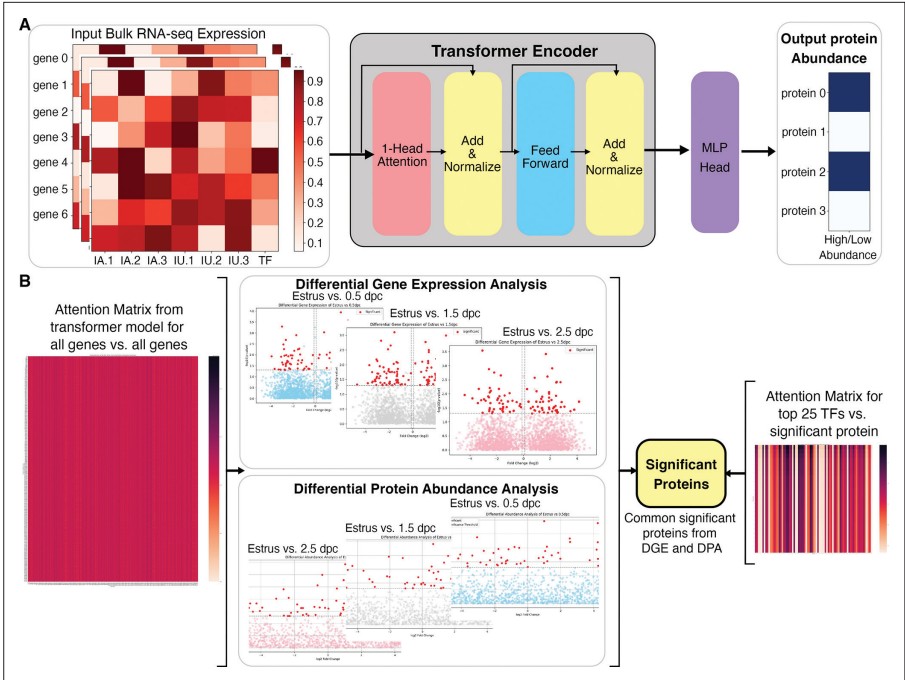

**Figure 4.** Overall architecture of the transformer-based model to predict proteomic abundance from bulk RNA-sequencing (RNA-seq) data of natural fertilization of oviduct. (**A**) Preprocessing steps using bulk RNA-seq count per million (cpm) normalization to calculate expression values. The transformer model is equipped with a single-layer transformer encoder featuring a single-head (1-Head) self-attention mechanism to predict the abundancy of proteins (abundant or not) from the input RNA-seq data. 'Head' refers to blocks, modules, or connections that perform specific tasks in neural networks. A specific threshold of 0.6/0.8 was defined to label proteins as high abundance or low abundance. The Multi-Layer Perceptron (MLP) Head refers to the output layer, which is designed to perform a classification task. In this model, The MLP layer uses a multi-layer perceptron or linear layer as the backbone to divide high abundance and low abundance based on the importance or attention weights given by the previous transformer layer. (**B**) The visual representation of a method to extract the top 25 TFs for differential significant proteins. DGE: differential gene expression, DPA: differential protein abundance, TF: transcription factor.

tool can be adapted to other biological disciplines to identify influential ITs and IPs using existing bulk RNA-seq databases. Overall, our study lays the groundwork for developing a novel and comprehensive AI model approach specifically designed to combine and predict influential ITs and IPs in biological samples.

## Evaluation of human hydrosalpinx Fallopian tubes compared to sperm-induced inflammation genes

To determine whether sperm-induced inflammatory responses in the mouse oviduct are similar to or different from human inflammation conditions, we reanalyzed publicly available scRNA-seq data from hydrosalpinx samples by *Ulrich et al., 2022*. We found that some of the sperm-induced inflammatory genes identified from our mouse study were present and upregulated in hydrosalpinx samples compared to healthy subjects (*Figure 5A*). However, the differentially expressed levels, for example the *CCL3* gene, appeared to be marginal between healthy versus hydrosalpinx samples (*Figure 5B, C* and *Supplementary file 5*). Nevertheless, the top five most enriched GOBPs related to inflammatory responses were Regulation of Complement Activation, Positive Regulation of Macrophage Migration Inhibitory Factor Signaling Pathway, MHC Class II Protein Complex Assembly, Positive Regulation of NK Cell Chemotaxis, and Negative Regulation of Metallopeptidase Activity (*Figure 5D*). These GOBPs differed from those identified in sperm-exposed mouse oviducts at 0.5 dpc, which were enriched for neutrophil-related pathways, unlike macrophages or NK cells in hydrosalpinx samples.

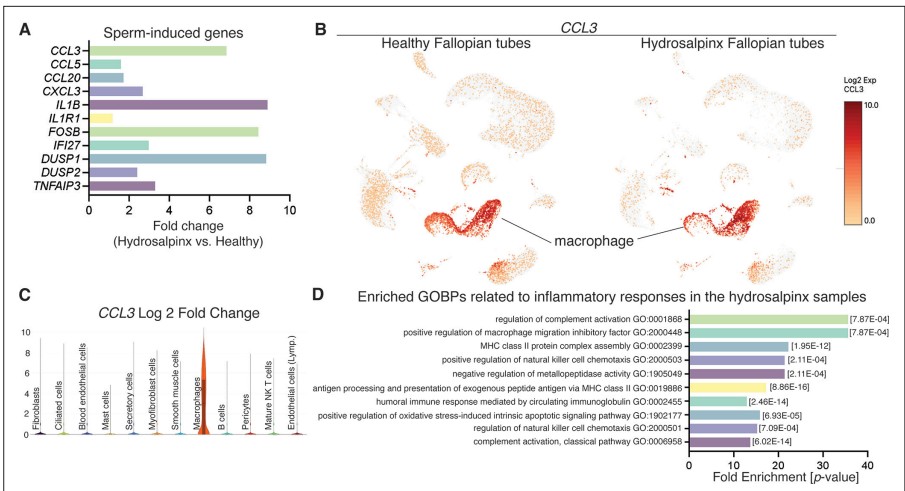

**Figure 5.** Reanalysis of data from *Ulrich et al., 2022* using hydrosalpinx versus healthy Fallopian tube samples from GSE178101. (**A**) Expression of sperm-induced genes identified from this current study in the hydrosalpinx compared to healthy Fallopian tube samples. (**B**) Uniform manifold approximation and projection (UMAP) of *CCL3* in healthy and hydrosalpinx Fallopian tubes in the macrophage populations. (**C**) Log2foldchange of *CCL3* in a violin plot comparing hydrosalpinx versus healthy Fallopian tubes. (**D**) Enriched gene ontology biological processes (GOBPs) related to inflammatory responses in the hydrosalpinx samples.

# Discussion

Here, we performed the in vivo multi-omics characterization of the oviduct in the mouse model (*Figure 6*). We integrated a total of 68 biological samples using bulk RNA-seq (24 total biological replicates), scRNA-sequencing (20 total biological replicates), and LC–MS/MS (24 total biological replicates) analyses. In addition, our bulk RNA-seq and proteomic data are immediately available to the scientific community in a web search format (details in Methods). Here, we reinforced significantly enriched pathways shared between different multi-omics techniques. We validated previous findings (*Harwalkar et al., 2021*) that the transcriptional profile of the oviduct between the IA and IU regions is unique and region-specific based on PCA analyses. Both the IA and IU regions are most distinct at 0.5 dpc compared to other timepoints based on PCA analyses, with unique transcription patterns becoming most disrupted at 0.5 dpc in both pregnancy and pseudopregnancy datasets. Large sets of DEGs display a dramatic shift from either being up- or downregulated between 0.5 and 1.5 dpc in all -omics characterizations. The changes at 0.5 dpc appear to subside at 1.5–3.5 dpc, with fewer unique clusters of genes that were dynamic between timepoints, indicating that either the absence of sperm or the presence of embryos drives the oviduct transition. However, the number of dynamic clusters of genes was greater in the IU region than in the IA region. This finding suggests that the IU region is more dynamic and responsive to the presence of gametes (sperm and oocytes/embryos) compared to the IA region.

At 0.5 dpc, we found that there were unique upregulated DEGs that corresponded with BPs involved in tissue remodeling and muscle filament sliding, such as ECM and collagen fibril organization. Wang and Larina showed that during this timepoint, ciliated epithelial cells in the ampulla region are responsible for creating a circular motion of the cumulus–oocyte complexes (COCs) within the ampulla (*Wang and Larina, 2021*). Here, using scRNA-seq analysis, we found that the ciliated cell population showed suppressed expression of genes involved in cilia assembly at SO 0.5 dpc and in SO Estrus (COCs are present in these two groups) in the IA region. This finding suggests that when the COCs are present in the ampulla, ciliated cells are functionally active. As SO increases the number of mature follicles (therefore, estrogen levels), ovulated eggs, and follicular fluid, it is also likely that these biological alterations could lead to changes in the protein abundance in the oviduct. Interestingly, the *Ephx2+* cluster was mainly present in the SO 0.5 dpc and SO estrus samples. *Ephx2* encodes epoxide hydrolase 2, which converts epoxides to dihydrodiols. Recent findings suggest that EPHX2 may play a role in primary hypertension in humans (*Ma et al., 2020*). However, the reproductive-related functions of EPHX2 have not yet been investigated. Therefore, we believe this presents an

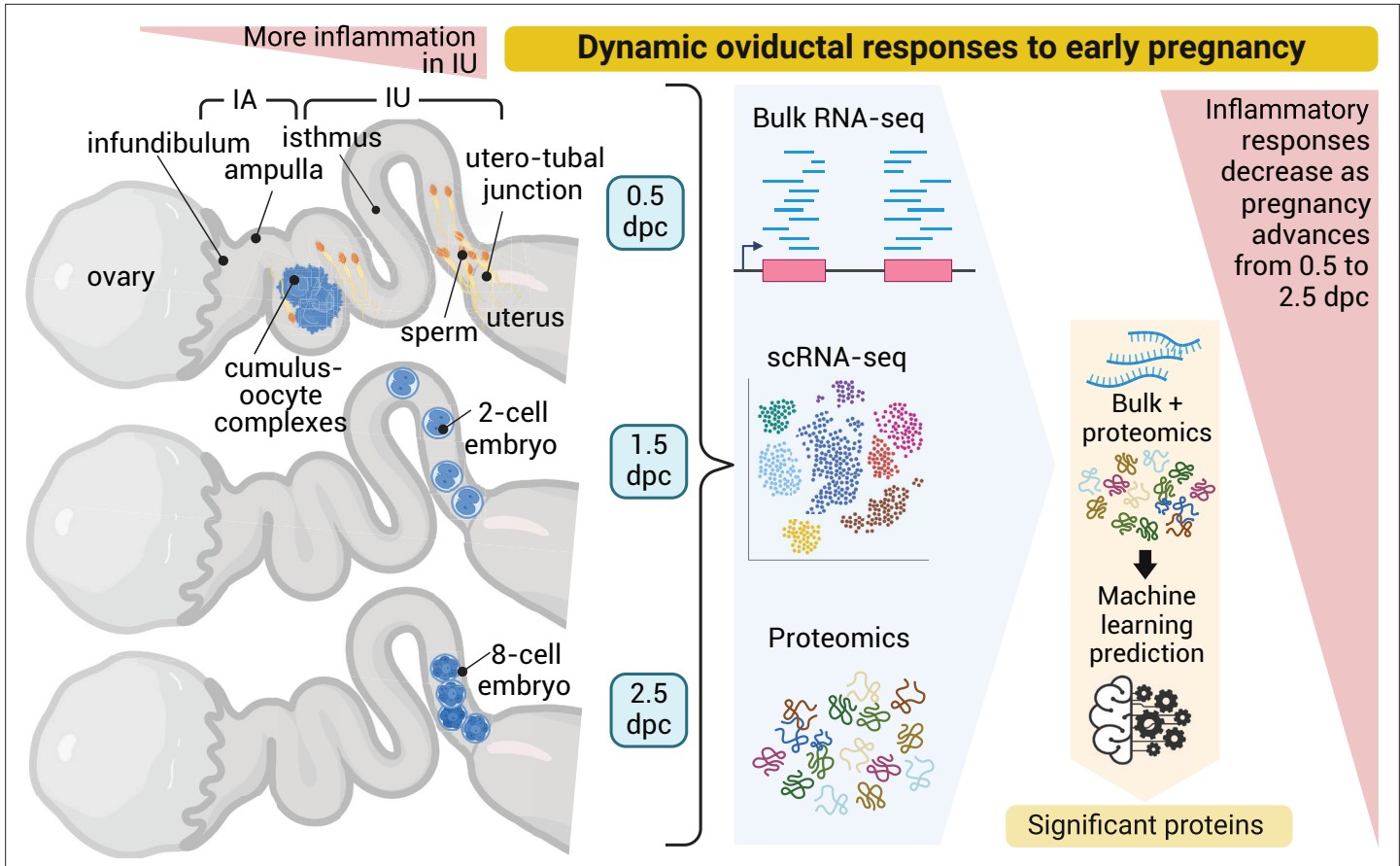

**Figure 6.** Dynamic oviductal response to early pregnancy in mice. Bulk RNA, scRNA-sequencing, and LC–MS/MS analyses were performed using the oviductal samples collected from mice at 0.5 days post coitus (dpc) through 2.5 dpc. Data were integrated and processed using machine learning to identify potential proteins crucial during each stage of early pregnancy. Created using BioRender.

opportunity for future research to define the role of *Ephx2* in the oviduct in response to SO during preimplantation embryo development.

The presence of sperm at 0.5 dpc strongly perturbed the IU region at 0.5 dpc, most likely due to a greater population of sperm in the IU, as a sperm reservoir (*Demott and Suarez, 1992*; *Suarez, 1998*), compared to the IA region. This perturbation was minimally detected at 0.5 dpp. Multi-omics analysis and observations in bulk RNA-seq, scRNA-seq, and luminal proteomics datasets are in agreement with the previous finding (*Bromfield et al., 2014*) that seminal fluid and sperm may be the dominant influencers for stimulating inflammatory responsive pathways in the oviduct at 0.5 dpc. We also established here, for the first time, that these observed inflammatory responses may be facilitated by the secretory cell population in the IU region when compared to other cell types. DUSP proteins modulate inflammation and antimicrobial immune responses (*Arthur and Ley, 2013*), and MAPK signaling pathways are involved in both pro- and anti-inflammatory pathways (*Arthur and Ley, 2013*; *Kaminska, 2005*). Therefore, we hypothesized that the observed inflammatory response was facilitated in part by the activation of MAPK signaling pathways, as indicated by a significant increase in expression of *Dusp5*, which was unique to the IU region after sperm exposure. Overall, the presence of sperm at 0.5 dpc induces a strong pro-inflammatory response in the IU region with upregulation of genes involved in inflammatory cytokines, neutrophil activation, lymphocyte recruitment, and T-cell proliferation. scRNA-seq data suggest that the oviduct is immunodynamic as indicated by the presence of immune cells as indicated by several immune markers such as neutrophils (*Ly6g+*), leukocyte (*Ptrpc+*), T cells (*Cd3d+*, *Cd3g+*), NK cells (*Nkg7+*, *Klrb1c+*), among others. This finding agrees with previous studies in human Fallopian tubes, as well as from our and other laboratories (*McGlade et al., 2021*; *Hu et al., 2020*; *Givan et al., 1997*).

Sperm migration from the uterus through the UTJ into the oviduct has been an observable phenomenon dating back five decades (*Krzanowska, 1974*). Additionally, phagocytic bodies engulfing sperm in mice luminal fluid in the isthmus region have also been observed in literature pre-dating the 1980s (*Chakraborty and Nelson, 1975*). The prevailing theory is that 'fit' sperm display inherently, via intracellular processes and genetic cargos, membrane 'passport' proteins that allow them to not only gain access through the UTJ, but also subsequently bind to the epithelium in the isthmus region (*Xiong et al., 2019*). The in situ hybridization analysis of the IU region reinforces these observations, suggesting additionally that not only do sperm require specific membrane proteins to function properly in the uterus and oviduct, but also that sperm must evade phagocytosis from an innate immune response. Our findings showed that *Ptprc*+ cells were present in the stromal and epithelial layers in the presence of sperm at 0.5 dpc in the UTJ. Similarly, a significant increase in *Tlr2*+ cells was observed at the epithelial lining of the isthmus and UTJ regions. *Tlr2* is a part of the Toll-like receptor superfamily of proteins that participate in and modulate immune responses (*Oliveira-Nascimento et al., 2012*). Previous and ongoing studies suggest an additional role for *Tlr2* in facilitating epithelial cell barrier integrity and remodeling after a significant immune response has occurred (*Abreu, 2010*; *Al Sadi et al., 2021a*; *Al Sadi et al., 2021b*; *Rakoff-Nahoum et al., 2004*). As such, we suggest a model where *Tlr2* expression increased at 0.5 dpc in response to the presence of sperm that may modulate epithelial cell integrity, thereafter, inducing remodeling in damaged cells at 1.5 and 2.5 dpc. Future studies need to be conducted to further reinforce this hypothesis.

Further indications of a pro-inflammatory condition induced by sperm at 0.5 dpc followed by epithelial cell remodeling at 1.5 dpc were observed in our luminal proteomics data. We observed an increase in NFκB fluorescent signal at 0.5 dpc, indicating conditions favorable for pro-inflammation. Previous work both in vivo and in vitro in the uterus and oviduct, respectively, indicate sperm have the capacity to induce immune-related responses (*Yousef et al., 2016*; *Marey et al., 2020*; *Schjenken et al., 2021*). In the uterus, observations suggest a hostile, phagocytic environment to remove excessive and dead sperm (*Marey et al., 2020*). Our findings suggest an equally hostile response to allogenic sperm in the oviduct at 0.5 dpc. However, this finding is in conflict with prior in vitro studies in the bovine oviductal epithelial cell (BOEC) culture model, in which sperm bind and induce anti-inflammatory cytokines, such as *TGFB1* (transforming growth factor β1) and *IL10*, while decreasing pro-inflammatory transcripts such as *TNFα* (tumor necrosis factor α) and *IL1B* in the BOECs (*Yousef et al., 2016*; *Marey et al., 2020*). It is possible that this discrepancy could be due to differences between (1) in vivo versus in vitro models and (2) murine versus bovine model organisms. Surprisingly, our luminal proteomics data suggest an exacerbated pro-inflammatory state in the SO condition, inducing greater dysregulation of pro-inflammatory pathways and epithelial cell remodeling.

At 1.5–3.5 dpc, oviductal transcriptional profiles were similar to each other compared to that of 0.5 dpc. During this preimplantation developmental period (1.5–3.5 dpc), embryos transit from the IA to the IU region (*Wang and Larina, 2021*; *Flores et al., 2020*). It indicates that there could be a critical transition of transcripts from 0.5 dpc to other timepoints when the embryos are present at 1.5–3.5 dpc. Therefore, our observations suggest that the oviduct provides an adaptive response in a unique manner during fertilization/preimplantation development, facilitating dynamic selection processes in the presence of gametes and embryos. At 1.5 dpc, 2-cell embryos were in the isthmus region. All embryos at later developmental stages 1.5–2.5 dpc were stalled in the lower isthmus and subsequently the UTJ region between 2.5 and 3.0 dpc. At 3.5 dpc, all embryos have transited from the oviduct to the uterus. Nutrients such as pyruvate, lactate, and amino acids are present in the oviductal fluid in several mammalian species (*Nieder and Corder, 1982*; *Tay et al., 1997*; *Nichol et al., 1992*). After fertilization, zygotes acquire pyruvate and lactate for their energy source (*Folmes and Terzic, 2014*). Then, the metabolism profile shifts from oxidative to glycolytic metabolism at later stages of preimplantation development (*Gardner et al., 1996*; *Absalón-Medina et al., 2014*). Here, we found that upregulated DEGs at 0.5 dpc were enriched for several energy metabolism BPs, including pyruvate metabolic, glucose catabolic process to pyruvate, canonical glycolysis, and glycolytic process through glucose-6-phosphose. These pathways are subsequently downregulated between 1.5 and 3.5 dpc. We showed that genes involved in these pathways were unique to the IU region with respect to differential expression analysis. Therefore, it is possible that the IU region is priming the environment to adjust to produce specific energy sources required for early and late embryo metabolism as the

embryo switches from utilizing pyruvate to utilizing glucose during successive developmental periods in the oviduct.

Lastly, we found that sperm-induced inflammatory conditions in the oviduct were potentially different than those of chronic inflammatory conditions in human Fallopian tubes. The inflammatory responses observed in mice and humans exhibited significant distinction based on immune cell involvement, mechanisms, and context. In mice, acute inflammation after sperm exposure could be primarily characterized by the activation of neutrophils, which serve as the first responders to injury or foreign bodies. In contrast, human Fallopian tubes with hydrosalpinx conditions displayed chronic inflammatory conditions predominantly involving macrophages and NK cells, suggesting a more complex and sustained immune response. It is also possible that inflammation in the oviduct differs between mice and humans. Understanding these species-specific variations is crucial for developing effective therapeutic strategies, as findings from murine models may not accurately translate to human inflammatory conditions due to the distinct immune dynamics at play.

In conclusion, we have demonstrated through a comprehensive multi-omics study of the oviduct that the transcriptomic and proteomic landscape of the oviduct at four different preimplantation periods was dynamic during natural fertilization, pseudopregnancy, and SO using three independent cell/tissue isolation and analytical techniques. Most novel findings from this study suggest that: (1) sperm were likely the key mediators in modulating inflammatory responses in the oviduct, potentially priming the oviduct to become tolerable to the presence of embryos, (2) inflammatory cytokine-mediated signals observed were more robustly amplified by the secretory epithelial cells of the oviduct, (3) the oviduct is an immuno-dynamic organ, alternating between a pro-inflammatory condition at 0.5 dpc to seemingly prioritizing epithelial barrier integrity/rejuvenation and cellular homeostasis between 1.5 and 2.5 dpc, and (4) the oviduct could provide necessary nutrient enrichment in the luminal fluid at different stages of embryonic development. In addition, an initial stage AI learning predictive model has been used to identify influential transcription factors and correlate predictive protein expressions. This initial AI model has recapitulated a similar progressive prediction of TFs correlating to IPs suggesting similar biological/cellular processes as our empirical in vivo multi-omics analysis. Overall, our findings reveal an adaptive oviduct with unique transcriptomic profiles in different oviductal regions, along with dynamic proteomics that may be specialized to influence sperm migration, fertilization, embryo transport, and development. These findings and techniques could facilitate developments to ensure a proper microenvironment for embryo development in vitro, assisting in establishing standard protocols at the laboratory, agricultural, and clinical levels.

## Materials and methods

### Animals

All animals were maintained at Washington State University and the University of Missouri and were handled according to Animal Care and Use Committee guidelines using approved protocols 6147, 6151, 38927, and 38961. ARRIVE guidelines have been followed for experiments in mice. C57BL/6J mice from Jackson Laboratories (Bar Harbor, ME) were used in this study. In all experiments, adult C57BL/6J female mice between 8 and 16 weeks were used. Mice were randomly assigned into groups. Based on our previous work, the sample size of at least three to four mice will provide 80% statistical power if there is at least a 20% difference between experimental groups when the statistical significance is considered at p < 0.05. Some females were naturally mated with fertile C57BL/6J males. Pseudopregnancy was induced by mating females with vasectomized males. The presence of a copulatory plug the next morning was considered 0.5 days post coitus (dpc) for females mated with fertile males and 0.5 days of pseudopregnancy (dpp) for females mated with vasectomized males.

### Hematoxylin and eosin staining

Oviductal tissues were dissected from 0.5 to 3.5 dpc/dpp of natural fertilization and pseudopregnancy, respectively. Oviducts were placed in cassettes individually and submerged in 10% formalin for 12–16 hr where they were then placed for storage in 70% ethanol the next day at 4°C. Tissue samples were then paraffin-embedded and sectioned at a 5-μm thickness. Sections were stained with H&E using a standard staining procedure as previously described (*McGlade et al., 2021*).

### Tissue collection for bulk RNA-seq

Oviductal tissues were collected and stored in pairs (one pair of oviducts per animal) at 0.5, 1.5, 2.5, and 3.5 dpc/dpp of natural fertilization and pseudopregnancy. For 0.5 dpc/dpp tissue, female mice were placed for mating at 21:00 hr. For 1.5, 2.5, and 3.5 dpc/dpp, female mice were placed for mating between 5 and 6 p.m. Oviducts were dissected and kept in 1 ml Leibovitz-15 (L15, Gibco, 41300070, Thermo Fisher Scientific, Carlsbad, CA) + 1% fetal bovine serum (FBS, Avantor 97068-091, Radnor Township, PA) media for transportation. Before sectioning the oviduct into two regions infundibulum + ampulla (IA), isthmus + uterotubal junction (IU), oviducts were flushed with L15 + 1% FBS media under a 37°C dissecting microscope (Leica MZ10f, Leica Microsystems, Buffalo Grove, IL). The presence of a minimum of 6 embryos per female was confirmed as a benchmark to represent the average litter size. Additionally, embryos were confirmed to be in the correct developmental stage and location in oviductal tissue samples. Then, oviducts were sectioned into IA and IU regions. We defined the IA region by including the infundibulum and cutting at turn three of the oviductal coil (the end of ampulla) (*Harwalkar et al., 2021*). Turn four to eleven was considered the IU region, which was stripped of uterine tissue enveloping the colliculus tubaris of the UTJ region (*Harwalkar et al., 2021*). Tissue samples were placed in a sterile Eppendorf tube and flash-frozen in liquid N$_2$. Samples were stored at −80°C for later RNA extraction. All dissections took place between 10:00 and 13:00 hr to decrease sample variation. The average time from cervical dislocation of the mouse to flash-freezing tissues was 15:43 (min:s). The oviducts were collected at the same timepoints as their dpc counterparts for DPP samples.

### Bulk RNA isolation, sequencing, and analysis

Both tissue and embryo total RNA were extracted utilizing the RNeasy Micro Kit (QIAGEN, Germantown, MD) according to the manufacturer's instructions. DNA digestion was performed with all samples using a QIAGEN RNase-free DNase Set (1500 K units). RNA was then shipped to the University of California San Diego (UCSD) for quality control, library preparation, and sequencing. RNA integrity (RIN) was verified using TapeStation for a minimum RIN value of 7. RNA from this study has an average RIN of 9.04. RNA libraries were prepared using the Illumina Stranded mRNA Prep kit (Illumina Inc, San Diego, CA). Then libraries were sequenced using the Illumina NovaSeqS4 platform with a read depth of 25 M reads/sample (*n* = 3/region/timepoint), paired-end, and 100 bp read length. FASTQ files were then analyzed utilizing BioJupies (*Torre et al., 2018*) and an integrated web application for differential gene expression and pathway analysis (iDEP) (*Ge et al., 2018*). The quality control, sequence alignment, quantification, DEG, heatmaps, and pathway and enrichment analyses were performed using default settings as indicated in BioJupies and iDEP web tools (*Torre et al., 2018*; *Ge et al., 2018*). In brief, FASTQ files were pseudoaligned, and DEGs were determined using DESeq. DEGs were then plotted as PCA and heatmaps through BioJupies. In some cases, read counts or reads per kilobase of transcript per million mapped reads (FPKM) were exported from BioJupies and imported into iDEP.92 for further pathway and KEGG analyses. InteractiVenn was used to generate common/overlap gene lists between different regions and timepoints. To validate that our isolation method and RNA-seq data analysis pipeline are reproducible with the previous report (*Roberson et al., 2021*) we evaluated the gene expression profiles of IA and IU regions from estrus samples (*n* = 3 mice/region). In agreement with the previous findings (*Roberson et al., 2021*), PCA plots showed that the IA and IU regions segregate from each other along the PC1 axis (74.3%) with respect to estrus (data not shown). Similar to the previous report, there was a significant indication of a region-specific expression of large subsets of genes.

### Single-cell isolation, library preparation, and single-cell RNA-seq

Another set of mice was used for single-cell isolations and scRNA-seq analysis. Mating and tissue collection protocols were similar to bulk RNA isolation described above, with the exception that female mice were superovulated using the protocol described previously (*Winuthayanon et al., 2015*) to ensure sufficient numbers of female mice at each timepoint could be harvested for single-cell isolation and library preparation within the same day (*n* = 3–4 mice/group). SO was performed by intraperitoneal injection of 5 IU pregnant mare serum gonadotropin (PMSG, Prospect HOR-272, East Brunswick, NJ). Forty-eight hours after PMSG injection, females were injected with 5 IU of human chorionic gonadotropin (hCG, Prospect HOR-250). Immediately after the hCG injection, females were

placed in fertile male cages for mating. Oviducts were collected at 0.5, 1.5, and 2.5 dpc and dissected into IA and IU regions before single-cell isolation. For the control group, oviducts were collected 16 hr post hCG injection. Trypsin-EDTA (0.25%, MilliporeSigma, T4049) was used for oviductal cell isolation using our previously described method (*McGlade et al., 2021*). The final cell concentration was targeted for 8000 cells/run. Cell singlets were captured for the library preparation using 10X Chromium Controller and Chromium Next GEM Single Cell 3′ GEM, Library & Gel Bead Kit v2 (10X Genomics, Pleasanton, CA). Libraries generated were then evaluated for quality using Fragment Analyzer (Agilent, Santa Clara, CA). Libraries were sequenced using Illumina HiSeq4000 at the University of Oregon, targeting 400 million reads/run, paired-end, and 100 bp read length. scRNA-seq web summary output for each dataset is listed in *Supplementary file 6a*.

## scRNA-seq analysis

Scanpy was used to analyze the scRNA-seq data. The generated loom files were read in as anndata objects and concatenated into a master anndata object. Preprocessing and quality control were performed similarly to the methods described in Scanpys clustering tutorial (*Wolf et al., 2018*) and Seurat's clustering tutorial (*Stuart et al., 2019*). Filtered out were cells expressing fewer than 200 genes, genes expressed in fewer than 3 cells, doublets (cells/droplets with counts for greater than 4000 genes), and cells with greater than 5% mitochondrial gene counts. Total counts were normalized to 10,000 for every cell, and log transformed. Highly variable genes were then identified using scanpy's 'highly_variable_genes' function with default parameters. Effects of mitochondrial gene expression and total counts were regressed out, and the data were scaled to unit variance and a mean of zero. Dimensionality reduction was first achieved through PCA with Scanpy's default parameters. To achieve further dimensionality reduction, a neighborhood graph of cells was computed, utilizing the top 40 principal components (PCs) and a neighborhood size of 10, then embedded utilizing uniform manifold approximation and projection, using the default parameters in Scanpy. Clustering of cells was achieved through Leiden clustering at a resolution of 0.1. Established marker genes were used to identify clusters as specific cell types: pan-epithelial (*Epcam+*), secretory (*Pax8+*), ciliated (*Foxj1+*), leukocytes (*Ptprc+*), antigen-presenting cells (*Cd74+*), monocytes and macrophages (*Ms4a7+Cd14+*), T-cells (*Cd3d+*, *Cd3g*), natural killer and NKT cells (*Nkg7+*, *Klrb1c+*), B-cells (*Cd79a+*, *Cd79b+*), granulocytes (*S100a8+*, *S100a9+*), neutrophils (*Ly6g+*), fibroblasts and stromal (*Pdgfra+*, *Twist2+*, *Dcn+*, *Col1a1+*), and endothelial (*Pecam1+*). Subsets containing only specific cell types (e.g., secretory cells), treatments (e.g., control, 0.5, 1.5, and 2.5 dpc), or regions (e.g., IA and IU) were created for specific downstream analyses and analyzed through the same process as above with identical parameters.

## Oviductal luminal fluid collection for luminal proteomic characterization

Oviducts were collected as pairs at estrus, 0.5, 1.5, and 2.5 dpc/SO of natural fertilization and superovulated fertilization, respectively. The estrus stage was determined by performing a vaginal lavage, followed by H&E staining. Datasets from the natural cycle and SO allowed us to directly compare the impact of exogenous hormone treatments on protein abundance and profile distinct from the physiological levels of hormones. In this context, our SO approach facilitates multi-dimensional analysis comparisons among naturally cycling bulk RNA-seq, SO scRNA-seq, and natural luminal proteomic biological replicates, enhancing confidence between different methods. This experimental design also reflects adaptive responses in the oviduct during natural fertilization and preimplantation development, influenced by PMSG and hCG treatments at both RNA and protein levels. Furthermore, SO is commonly used in female reproduction to synchronize estrus cycles in animals, thus reducing variables at each collection timepoint.

For estrus SO, oviducts were collected the day after hCG injections between 10:00 and 13:00 hr. The presence of cumulus mass cells containing oocytes was also confirmed. For 0.5 dpc/SO tissue, female mice were placed for mating at 9 p.m. For 1.5 and 2.5 dpc/SO, female mice were placed for mating between 5 and 6 p.m. Oviducts were dissected and washed in a petri dish containing a 25-µl drop of phosphate-buffered saline (PBS) + HALT (1×) (Thermo Scientific, 78440). Once transported to a dissection scope, oviducts were then moved to a fresh adjacent 25 µl drop of PBS + HALT (1×). Inserting a dulled 30 G needle and syringe into the UTJ, each oviduct was subsequently flushed with 100 µl PBS + HALT (1×), for a total sample volume of 225 µl. Next, we observed, staged, and removed oocytes/embryos present in the sample drop via mouth pipetting ensuring to take the least amount

of sample fluid possible. The presence of a minimum of 6 embryos per female was confirmed as a benchmark to represent the average litter size. Once oocytes/embryos were removed, we placed the sample drop in a 1.5-ml Eppendorf tube and centrifuged at 2200 × *g* for 15 min to remove any additional cell debris or blood cells that may be present after flushing. The supernatant was removed, and we performed additional centrifugation at 5000 × *g* for 10 min. Once centrifugation was complete, the supernatant was placed/pooled and flash frozen with liquid $N_2$. Pooled samples (*n* = 3 mice/timepoints) were stored at −80°C. Every sample submitted for LC–MS/MS contained five paired oviduct flushes at each respective timepoint/condition. The average collection time from cervical dislocation to flushing was 10:47 (min:s) before subsequent centrifugations. Once all samples were collected, they were shipped on dry ice overnight to Tymora Analytical Operations (West Lafayette, IN) to perform LC–MS/MS analysis.

## ELISA analysis

Enzyme-linked immunoassay (ELISA) was utilized to establish in vivo translation of pro-inflammatory cytokine IL1β. Pairs of oviduct tissue from each biological replicate from the IU region were frozen individually, and subsequent protein extraction/tissue disruption of a single IU oviduct at each timepoint/condition was utilized. 2.5 µg total protein concentration from each biological replicate was administered in the assay. Three technical replicates per individual biological replicate at each timepoint/condition were analyzed. Oviductal tissues were collected as pairs (one pair of oviducts per animal) at 0.5 and 1.5 dpc/dpp of natural fertilization and pseudopregnancy, respectively. Oviducts were dissected and kept in 1 ml Leibovitz-15 (L15, Gibco, 41300070, Thermo Fisher Scientific) + 1% FBS (Avantor 97068-091, Radnor Township, PA) media for transportation. Before sectioning the oviduct into two regions (IA and IU), oviducts were flushed with L15 + 1% FBS media under a 37°C dissecting microscope (Leica MZ10f, Leica Microsystems, Buffalo Grove, IL). The presence of a minimum of 6 embryos per female was confirmed as a benchmark to represent the average litter size. Additionally, embryos were confirmed to be in the correct developmental stage and location in oviductal tissue samples. Oviducts were sectioned into IA and IU regions. Tissue samples were stored individually (the pair of oviducts were stored individually) in two separate sterile Eppendorf tubes and flash-frozen in liquid $N_2$. Samples were stored at −80°C for later protein extraction (TPER Tissue Protein Extraction Reagent (Thermo Scientific: 78510) + HALT (1×)). IL1β ELISA (ab197742, abcam, Waltham, MA) was performed according to the manufacturer's protocol.

## NFκB immunofluorescent staining

NFκB immunofluorescent staining was performed to evaluate the degree of inflammation activation in oviductal cells during fertilization. Following dissection, oviducts were placed in cassettes individually and submerged in 10% formalin for 12–16 hr where they were then placed for storage in 70% ethanol the next day at 4°C. They were subsequently processed and embedded in paraffin wax. In short, oviductal tissues were sectioned to 5 µm with a microtome. Sectioned samples were placed on Superfrost Plus Slides and baked overnight on a heat plate at 37°C. Slides were processed in xylene, followed by an alcohol series (100%, 95%, and 70%). Antigen retrieval was performed with sodium citrate retrieval buffer + 0.05% Tween-20 (pH 6.0) in a pressure cooker at 90°C for 10 min. Slides were rinsed with 1× TBST (0.05% Tween-20) and blocked with 1× TBST + 5% normal goat serum (NGS) cocktail for 60 min at RT before applying the NFκB primary antibody (Cell Signaling, 6956, RRID:AB_10828935, 1:1000) in 1× TBST cocktail containing 1% bovine serum albumin (BSA) overnight (12–16 hr) in a 1× TBST humidified chamber at 4°C. Slides were washed the next morning with 1× TBST before the secondary (1:1500) antibody (Jackson Immuno Research, 115-585-146) 1× TBST + 1% NGS cocktail was applied for 1 hr, covered from light, at RT. Slides were rinsed with 1× TBST before ProLong Diamond Antifade Mounting agent with DAPI (Invitrogen, P36962) was applied. The stained sections were subsequently covered with a glass coverslip. Stained slides were placed at 4°C for at least 24 hr before imaging immunofluorescence using a light microscope (Leica DMi8, Leica Microsystems). To establish relatively quantitative significance, 15 measurements were taken across two stained representative images from 20× objectives using FIJI software, for a total of 30 measurements per timepoint/condition for relative fluorescent strength.

## RNA in situ hybridization

To perform in situ hybridization, oviductal tissues were dissected as pairs (one pair of oviducts per animal) from 0.5 to 1.5 dpc/dpp of natural fertilization and pseudopregnancy, respectively. Oviducts were placed in cassettes individually and submerged in 10% formalin for 12–16 hr where they were then placed for storage in 70% ethanol the next day at 4°C. Tissue samples were then paraffin-embedded and sectioned at a 5-μm thickness, where subsequent staining of target RNAs was performed utilizing ACDbio RNAscope Multiplex Fluorescent Reagent Kit V2 in accordance with ACDbio recommended protocols. RNAscope probes used were as follows: #317521-*Tlr2*-C1, #506391-*Ly6g*-C2, and #318651-*Ptprc*-C3. Images were taken using a Leica DMi8 light microscope with a K8 camera (Leica Microsystems). Three technical replicates per individual biological replicate at each timepoint/condition were analyzed utilizing ImageJ (FIJI) color histogram quantification software, followed by GraphPad and two-way ANOVA statistical analysis comparing the mean of every row (gene target) to every column (timepoint/condition) to establish significance.

## p38 and phosphorylated-p38 immunoblotting

Immunoblotting was used to establish in vivo expression and phosphorylation of p38. Pairs of oviduct tissue from each biological replicate from the IU region were frozen individually, and subsequent protein extraction/tissue disruption of a single IU oviduct at each timepoint/condition was utilized. 12 μg total protein concentration from each biological replicate was administered in the assay. Three individual biological replicates at each timepoint/condition were analyzed utilizing FIJI, GraphPad software, and two-way ANOVA statistical analysis was performed to establish significance. Oviductal tissues were collected as pairs at 0.5 and 1.5 dpc/dpp of natural fertilization and pseudopregnancy, as described above.

TPER + HALT (1×) cocktail (150 μl) was applied to frozen tissue IU samples, followed immediately by homogenization of cells via a tissue disruptor. Tissues were incubated in the TPER + HALT (1×) cocktail for 2 hr on ice and were vigorously vortexed every 30 min for approximately 10 s. HALT protease inhibitor was introduced again at the 1 hr incubation for a final concentration of 1× to ensure continuous inhibition of proteases. Homogenized tissue samples were pelleted at 6000 × $g$ for 5 min at 4°C, with the subsequent removal of the supernatant, which underwent an additional centrifugation treatment. 10 μl of supernatant was aliquoted out of the cell-debris purified supernatant for BCA protein concentration determination. The remaining supernatant (~140 μl) was flash frozen in liquid $N_2$ and stored at −80°C. Protein supernatants were incubated with 4× Laemmli buffer containing β-mercaptoethanol at a final concentration of 12 μg total protein and heated to 95°C for 7 min. Gel electrophoresis was performed with 1× running buffer (25 mM Tris, 192 mM glycine, 0.1% SDS) at 90 V constant for 10 min, then after increasing constant voltage to 120 V for approximately 1.5 hr. Polyvinylidene difluoride (Immobilon, IPVH00010) transfer membranes were incubated in methanol for 10 min and washed in 1× transfer buffer (25 mM Tris, 192 mM glycine) before transfer of proteins from Tris-glycine SDS–polyacrylamide gels. The transfer occurred on the ice at 90 V for approximately 1.5 hr. Then membranes were subsequently washed (3×, 5 min) with 1× PBS, 0.05% Tween-20 (PBST) at RT before being blocked with 5% non-fat milk (ChemCruz, sc-2325) in 1× PBST for 1.5 hr at RT. Transfer membranes were thereafter treated with a primary p38 (Cell Signaling, 9212S, RRID:AB_330713) (1:1000 dilution) or phosphorylated-p38 (Cell Signaling, 4511S, RRID:AB_2139682) primary antibody (1:1000 dilution) in 1× PBST + 1% BSA overnight at 4°C. Membranes were washed before the secondary goat anti-rabbit antibody (abcam, ab97051) 1× PBST + 1% non-fat milk cocktail was incubated at RT for 1.5 hr. Membranes were then incubated with Bio-Rad Clarity Western ECL substrate chemiluminescence kit (Bio-Rad, 170-5060) and subsequently imaged utilizing a Bio-Rad Molecular Imager ChemiDoc XRS+.

## Predictive transformer model for predicting proteomics data from transcriptomics data and identifying key transcription factors

The development of an Integrative AI model involves the utilization of a transformer encoder with a single-head self-attention mechanism. The model's architecture is depicted in *Figure 1*. Input data comprise bulk RNA-seq expressions obtained from naturally fertilized oviduct mice at one of various stages such as estrus, 0.5 dpc, 1.5 dpc, and 2.5 dpc, and the output is the abundance of proteins. Preprocessing steps were applied to raw reads from bulk RNA-seq and raw protein abundance values,

which involved removing genes and proteins lacking recorded expression and abundance values across all timepoints, respectively. Normalization techniques were employed on the data, including counts per million (CPM) normalization (*Johnson and Krishnan, 2022*) for bulk RNA counts to calculate expression values. Furthermore, these values underwent percentile normalization to be confined in the range [0–1], a critical step for machine learning models to manage exploding/vanishing gradients during training (*Huang et al., 2023*). Protein abundance values were normalized using log-min-max within each timepoint sample. A specific threshold of 0.6/0.8 was defined to label proteins as high abundance or low abundance. The bulk RNA-seq expression matrix, encompassing samples from IA and IU regions, along with an additional feature indicating if a gene is a transcription factor, was incorporated and randomly sampled for data augmentation. The transformer model is equipped with a single-layer transformer encoder featuring a single-head self-attention mechanism to predict the abundancy of proteins (abundant or not) from the input RNA-seq data. The attention mechanism directs its focus toward crucial segments of the input, capturing the key genes (e.g., transcription factors) that influence protein abundance. The augmented RNA-seq data from estrus, 0.5 dpc, and 1.5 dpc, and the corresponding extracted protein abundance labels, were used to train and validate the transformer model. Subsequently, the model's performance in predicting the abundance of proteins from RNA-seq data was blindly tested using samples from 2.5 dpc not used in the training and validation. Moreover, the attention matrix derived from the trained transformer model was checked against the results of the differential gene expression analysis and the differential protein abundance analysis to identify significant proteins and the key transcription factors that may influence them across different timepoints. The evaluation results of the model are shown in *Supplementary file 6b*.

## Enrichr pathway analysis: Reactome (2022) and Gene Ontology (GO Biological Processes 2023) bulk RNA and luminal proteomics analysis

DEG lists were generated for bulk RNA and luminal proteomic data analysis utilizing Biojupies differential expression software and Perseus software. Differentially abundant proteins were transformed with a Gaussian normal assumption before being subjected to two/single-tailed *t*-test statistical analysis in Perseus. For a greater description of this integration, refer to the respective methods section below. Differential gene/protein lists were generated with an FDR <0.05 before subsequent lists were submitted to Enrichr for biological pathway analysis. Reactome (2022) and GO Biological Processes 2023 tables were generated and utilized in combination as each database establishes pathway p-value significance differently.

## Gene ontology scRNA analysis

DEGs were identified using scanpy's 'highly_variable_genes' function with default parameters. Generated DEG sublists containing up- and downregulated genes with a $\log_2$FC $\geq 1$ or $\leq -1$, respectively, were then filtered for genes/proteins. The filtered gene/protein lists were submitted to Enrichr for Gene Ontology enrichment analysis. Exported data were plotted utilizing R studio via ggplot (*Wickham, 2016*), InteractiVenn, and Perseus software.

## Gaussian normal distribution assumption (continuous probability distribution)

The Gaussian normal distribution is a mathematical assumption. This mathematical assumption is based on the existence of a continuous random variable. We assume that any single empirically measured protein value (random variable) will not yield the same empirical measurement if subsequent repeated measurements are taken on the same sample. Therefore, we assume that any repeated empirical measurement of any one protein will adhere to a distribution rather than being an absolute measurement. We applied this assumption to our pooled oviductal luminal protein biological replicates to extend our analysis with respect to utilizing statistical tests to identify significantly altered protein abundances during preimplantation development. However, we limited our assumption range to 1 SD above and below all empirically measured protein values. Applying these parameters generates two additional artificial but probable values centered around the true empirical measurement. For example, statistical *t*-tests carried out with this integration will assign empirical measurements as the means for statistical comparisons. This transformation allowed for the establishment of significance between proteins with pooled (3 pairs of oviducts, for a total of 6

pooled oviducts per timepoint/condition) biological replicates at each unique timepoint/condition. Two/single-tailed *t*-tests and PCA were generated with Perseus software to establish significant differences. Significant differentially abundant proteins were assigned and visualized with a Venn diagram produced by the interactiVenn webpage, followed by Enrichr Reactome and GO biological processes analysis.

## Acknowledgements

The authors thank Kalli Stephens for helping maintain the C57BL/6J mouse colony and Gerardo Herrera for initial analysis of scRNA-seq data. This study is supported in part by the Eunice Kennedy Shriver National Institute of Child Health & Human Development, National Institutes of Health award numbers R01HD097087 to WW, National Science Foundation grants (DBI2308699 and CCF2343612) to JC, Washington State University (WSU) Office of Research (RA + $10K) Award to RMF, and WSU NIH Protein Biotechnology Training Grant (T32GM008336) to DJC.

## Additional information

### Funding

| Funder | Grant reference number | Author |
|---|---|---|
| Eunice Kennedy Shriver National Institute of Child Health and Human Development | R01HD097087 | Wipawee Winuthayanon |
| National Science Foundation | DBI2308699 | Jianlin Jack Cheng |
| National Science Foundation | CCF2343612 | Jianlin Jack Cheng |
| National Institute of General Medical Sciences | T32GM008336 | Daniel J Carulli |
| University of Missouri | Start-up fund | Wipawee Winuthayanon |
| Washington State University | RA+$10K | Ryan M Finnerty |

The funders had no role in study design, data collection and interpretation, or the decision to submit the work for publication.

### Author contributions

Ryan M Finnerty, Conceptualization, Data curation, Formal analysis, Validation, Investigation, Visualization, Methodology, Writing – original draft, Writing – review and editing; Daniel J Carulli, Formal analysis, Investigation, Visualization, Methodology, Writing – review and editing; Akshata Hedge, Yanli Wang, Frimpong Boadu, Data curation, Software, Formal analysis, Investigation, Visualization, Methodology, Writing – review and editing; Sarayut Winuthayanon, Resources, Software, Methodology, Writing – review and editing; Jianlin Jack Cheng, Conceptualization, Resources, Data curation, Software, Formal analysis, Supervision, Funding acquisition, Methodology, Writing – original draft, Writing – review and editing; Wipawee Winuthayanon, Conceptualization, Data curation, Formal analysis, Supervision, Funding acquisition, Validation, Investigation, Visualization, Methodology, Writing – original draft, Project administration, Writing – review and editing

### Author ORCIDs

Wipawee Winuthayanon  https://orcid.org/0000-0002-5196-8471

### Ethics

All animals were maintained at Washington State University and the University of Missouri and were handled according to Animal Care and Use Committee guidelines using approved protocols 6147, 6151, 38927, and 38961.

Reviewer #1 (Public review): https://doi.org/10.7554/eLife.100705.3.sa1
Reviewer #2 (Public review): https://doi.org/10.7554/eLife.100705.3.sa2
Author response https://doi.org/10.7554/eLife.100705.3.sa3

## Additional files

### Supplementary files

Supplementary file 1. Differential gene expression analysis from estrus. We first performed differential gene expression (DGE) in estrus and extracted significant proteins (protein-coding genes). Then, the significant proteins were used to extract top 25 transcription factors (TFs) influencing those proteins. For DEG, FDR threshold was set to <0.05 and $\log_2$foldchange of >0.1. Enrichr Reactome and GO biological process (BP) were used to determine the enriched pathways from the TFs and influenced proteins (IPs). Each excel file contains a table with Column names or header names being significant proteins in worksheet named 'AllTfs_diffcodinggene_sample'; rows 2–26 represent the top 25 transcription factors (TFs) influencing the corresponding protein abundance in row 1.

Supplementary file 2. Differential gene expression analysis between the estrus and 0.5 dpc. We first performed differential gene expression (DGE) between Control (Estrus) and timepoints and extracted significant proteins (protein-coding genes). Then, the significant proteins were used to extract top 25 transcription factors (TFs) influencing those proteins. For DEG, FDR threshold was set to <0.05 and $\log_2$foldchange of >0.1. Enrichr Reactome and GO biological process (BP) were used to determine the enriched pathways from the TFs and influenced proteins (IPs). Each excel file contains a table with Column names or header names being significant proteins in worksheet named 'AllTfs_diffcodinggene_sample'; rows 2–26 represent the top 25 transcription factors (TFs) influencing the corresponding protein abundance in row 1.

Supplementary file 3. Differential gene expression analysis between the estrus and 1.5 dpc. We first performed differential gene expression (DGE) between Control (Estrus) and timepoints and extracted significant proteins (protein-coding genes). Then, the significant proteins were used to extract top 25 transcription factors (TFs) influencing those proteins. For DEG, FDR threshold was set to <0.05 and $\log_2$foldchange of >0.1. Enrichr Reactome and GO biological process (BP) were used to determine the enriched pathways from the TFs and influenced proteins (IPs). Each excel file contains a table with Column names or header names being significant proteins in worksheet named 'AllTfs_diffcodinggene_sample'; rows 2–26 represent the top 25 transcription factors (TFs) influencing the corresponding protein abundance in row 1.

Supplementary file 4. Differential gene expression analysis between the estrus and 2.5 dpc. We first performed differential gene expression (DGE) between Control (Estrus) and timepoints and extracted significant proteins (protein-coding genes). Then, the significant proteins were used to extract top 25 transcription factors (TFs) influencing those proteins. For DEG, FDR threshold was set to <0.05 and $\log_2$foldchange of >0.1. Enrichr Reactome and GO biological process (BP) were used to determine the enriched pathways from the TFs and influenced proteins (IPs). Each excel file contains a table with Column names or header names being significant proteins in worksheet named 'AllTfs_diffcodinggene_sample'; rows 2–26 represent the top 25 transcription factors (TFs) influencing the corresponding protein abundance in row 1.

Supplementary file 5. Differential gene expression analysis between hydrosalpinx versus healthy Fallopian tubes from PMID: 35320732 (GSE178101). We first performed differential expressed genes using Loupe Browser. Then, the significant genes were used to determine the GO biological process (BP).

Supplementary file 6. scRNA-seq web output summary and predicted protein abundance by the transformer model. (A) scRNA-seq output for each dataset in this study. (B) The result of predicting protein abundance from bulk RNA-seq data at 2.5 dpc by the transformer model. The model is used to classify proteins into two categories, abundant or not abundant, according to two different thresholds: 0.6 and 0.8, respectively. The model can rather accurately predict the abundant proteins from RNA-seq data.

MDAR checklist

## Data availability

Raw data as fastq files were deposited at Gene Expression Omnibus (GSE270654). Bulk-RNA seq, scRNA-seq, and proteomic data are available in the web search format at https://genes.winuthayanon.com/winuthayanon/oviduct_bulkRNA-seq_pregnancy/, https://genesearch.org/winuthayanon/Oviduct_pregnancy/, and at https://genes.winuthayanon.com/winuthayanon/oviduct_proteins/, respectively.

The following dataset was generated:

| Author(s) | Year | Dataset title | Dataset URL | Database and Identifier |
|---|---|---|---|---|
| Finnerty RM, Carulli DJ, Hedge A, Wang Y, Baodu F, Winuthayanon S, Cheng J, Winuthayanon W | 2024 | Multi-omics analyses and machine learning prediction of oviductal responses in the presence of gametes and embryos | https://www.ncbi.nlm.nih.gov/geo/query/acc.cgi?acc=GSE270654 | NCBI Gene Expression Omnibus, GSE270654 |

The following previously published dataset was used:

| Author(s) | Year | Dataset title | Dataset URL | Database and Identifier |
|---|---|---|---|---|
| Ulrich ND, Shen YC, Ma Q, Yang K, Hannum DF, Jones A, Machlin J, Smith YR, Schon SB, Shikanov A, Marsh EE, Lieberman R, Gurczynski SJ, Moore BB, Jz Li, Hammoud S | 2022 | scRNA-seq analysis of hydrosalphinx disease state Vs healthy state of human fallopian tubes | https://www.ncbi.nlm.nih.gov/geo/query/acc.cgi?acc=GSE178101 | NCBI Gene Expression Omnibus, GSE178101 |

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
