## [Editor Report · eLife Assessment]

This **important** study reports the transcriptomic and proteomic landscapes of the oviducts at four different preimplantation stages during natural fertilization, pseudopregnancy, and superovulation. The supporting data are **convincing**. This work will be of interest to reproductive biologists and clinicians practicing reproductive medicine.

---

## [Referee Report · Reviewer #1 (Public review)]

Summary:

The paper demonstrated through a comprehensive multi-omics study of the oviduct that the transcriptomic and proteomic landscape of the oviduct at 4 different preimplantation periods was dynamic during natural fertilization, pseudopregnancy, and superovulation using three independent cell/tissue isolation and analytical techniques. This work is very important for understanding oviductal biology and physiology. In addition, the authors have made all the results available in a web search format, which will maximize the public's access and foster and accelerate research in the field.

Strengths:

(1) The manuscript addresses an important and interesting question in the field of reproduction: how does the oviduct at different regions adapt to the sperm and embryos for facilitating fertilization and preimplantation embryo development and transport?

(2) Authors used cutting-edge techniques: Integrated multi-modal datasets followed with in vivo confirmation and machine learning prediction.

(3) RNA-seq, scRNA-seq and proteomic results are immediately available to the scientific community in a web search format

(4) Substantiated results indicate the source of inflammatory responses was the secretory cell population in the IU region when compared to other cell types; sperm modulate inflammatory responses in the oviduct; the oviduct displays immuno-dynamism.

In addition, the revised version has addressed weaknesses adequately.

(1) The revised version provided a clear explanation and the rationale for using the superovulation model.

(2) The revised version generated a graphic abstract/summary of their major findings.

---

## [Referee Report · Reviewer #2 (Public review)]

The manuscript investigates oviductal responses to the presence of gametes and embryos using a multi-omics and machine learning-based approach. By applying RNA sequencing (RNA-seq), single-cell RNA sequencing (sc-RNA-seq), and proteomics, the authors identified distinct molecular signatures in different regions of the oviduct, proximal versus distal. The study revealed that sperm presence triggers an inflammatory response in the proximal oviduct, while embryo presence activates metabolic genes that provide nutrients to the developing embryos. Overall, this study offers valuable insights and will likely be of great interest to reproductive biologists and researchers in oviduct biology.

---

## [Author Response]

The following is the authors’ response to the original reviews.

**eLife Assessment**
This important study reports the transcriptomic and proteomic landscape of the oviducts at four different preimplantation periods during natural fertilization, pseudopregnancy, and superovulation. The data presented convincingly supported the conclusion in general, although more analyses would strengthen the conclusions drawn. This work will interest reproductive biologists and clinicians practicing reproductive medicine.

We appreciate the concise summary and agree that additional experiments can reinforce the fidelity of predictions made by our robust bioinformatic characterization of the oviduct. Our robust bioinformatic model appears reproducible as similar pathway trends have been produced in all three datasets, lending confidence for future researchers to establish testable hypotheses more effectively.

**Reviewer #1 (Public review):**
The paper demonstrated through a comprehensive multi-omics study of the oviduct that the transcriptomic and proteomic landscape of the oviduct at 4 different preimplantation periods was dynamic during natural fertilization, pseudopregnancy, and superovulation using three independent cell/tissue isolation and analytical techniques. This work is very important for understanding oviductal biology and physiology. In addition, the authors have made all the results available in a web search format, which will maximize the public's access and foster and accelerate research in the field.Strengths:(1) The manuscript addresses an important and interesting question in the field of reproduction: how does the oviduct at different regions adapt to the sperm and embryos for facilitating fertilization and preimplantation embryo development and transport?(2) Authors used cutting-edge techniques: Integrated multi-modal datasets followed by in vivo confirmation and machine learning prediction.(3) RNA-seq, scRNA-seq, and proteomic results are immediately available to the scientific community in a web search format.(4) Substantiated results indicate the source of inflammatory responses was the secretory cell population in the IU region when compared to other cell types; sperm modulate inflammatory responses in the oviduct; the oviduct displays immuno-dynamism.

We sincerely thank you for your thorough and insightful review of our manuscript. Your comprehensive summary accurately captures the essence of our multi-omics study on oviductal biology, highlighting its importance in understanding reproductive physiology. We are particularly grateful for your recognition of our study's strengths. In the revised manuscript, we have added another searchable scRNA-seq data on our public website; https://genesearch.org/winuthayanon/Oviduct_pregnancy/. We have also addressed the weaknesses in the response below in our revised manuscript.

Weaknesses:(1) The rationale for using the superovulation model is not clear. The oviductal response to sperm and embryos can be studied by comparing mating with normal and vasectomized mice and comparing pregnancy vs pseudopregnancy (induced by mating with vasectomized males). Superovulation causes supraphysiological hormone levels and other confounding conditions.

We agree with this assessment that superovulation changes the hormonal levels and could have a confounding impact on the oviduct function. As such, for all experiments involving pseudopregnant datasets, pseudopregnancy was induced by mating females with vasectomized males without superovulation. Our oviductal luminal protein content analysis was collected from oviductal fluid from pregnant females with and without superovulation. This allowed us to directly compare the impact of superovulation on protein abundance and profile. In the revised manuscript, we have provided clarifying statements on using superovulation in our Method section, which reads

“Datasets from the natural cycle and SO allowed us to directly compare the impact of exogenous hormone treatments on protein abundance and profile distinct from the physiological levels of hormones”.

One exception for using superovulation in the absence of a “natural mating” group for comparison is the scRNA-seq dataset. As single-cell libraries should be performed in a single run to avoid batch effects, we need to ensure that a sufficient number of females were pregnant for single-cell isolation (we used ~4 mice/time point). Therefore, superovulation was used to synchronize and ensure that the females were receptive to mating. At the time of our sample collection, single nuclei isolation methods (freeze tissue now, isolate nuclei later) had not been reliable or standardized. We tried to synchronize females using the male bedding without superovulation. However, we would still need to set up at least 12-15 females per pregnancy timepoint to mate with male mice, totaling ~48-60 mice each night. Due to budget constraints and vivarium space limitations, we were not able to do so. We have included a similar statement to clarify the justifications in the revised Methods, which reads,

“Mating and tissue collection protocols were similar to bulk RNA isolation described above, with the exception that female mice were superovulated using the protocol described previously (73) to ensure sufficient numbers of female mice at each timepoint could be harvested for single cell isolation and library preparation within the same day (n = 3-4 mice/group)”.

(2) This study involves a very complex dataset with three different models at four time points. If possible, it would be very informative to generate a graphic abstract/summary of their major findings in oviductal responses in different models and time points

Thank you for this suggestion. We have now included the graphical abstract to accompany our final version of the manuscript.

(3) The resolution of Figures 3A-3C in the submitted file was not high enough to assess the authors' conclusion.

We have now used a higher magnification of images in Figures 3A-C in the revised version.

(4) The authors need to double-check influential transcription factors identified by machine learning. Apparently, some of them (such as Anxa2, Ift88, Ccdc40) are not transcription factors at all.

We appreciate the recognition of this oversight. In the revised manuscript, we have clarified and stated the distinction between ‘influential transcripts’ and ‘influenced proteins’, which now reads,

“The top 25 “influential” transcripts (ITs) with the highest attention scores from all the transcription factors present in bulk RNA-seq data were extracted for every potentially influenced protein (IP) in the empirical proteomics datasets”.

**Recommendations for the authors:**
(1) What are the stained debris/nuclei surrounding oocytes/fertilized eggs in Figure 1A? Please indicate in figure legends.

We have edited Figure 1A with black arrows that highlight the stained cumulus cells surrounding the ovulated eggs/fertilized eggs, together with a revised Figure legend, which now reads, “Arrows indicate cumulus cells surrounding the eggs/fertilized eggs called cumulus-oocyte complexes”.

(2) "Then, oviducts were sectioned into IA and IU regions" The Ampulla region is quite a long tube. Could authors provide details about the cutting border between IA and IU regions?

We have now included a literature defining the number of turns in the coiled mouse oviduct and how we cut between the IA and IU regions in the Method section, which reads,

“We defined the IA region by including the infundibulum and cutting at turn three of the oviductal coil (the end of ampulla) (5). Turn four to eleven was considered the IU region, which was stripped of uterine tissue enveloping the colliculus tubaris of the UTJ region (5)”.

(3) "In this experiment, superovulation (SO) using exogenous gonadotropins was used due to technical limitations of sample collection for single-cell processing." It was not clear. What was the technical limitation of sample collections?

As indicated in response to the public review above, we have now clarified that we used superovulation for scRNA-seq analysis to ensure that a sufficient number of females were pregnant for singlecell isolation (we used ~4 mice/time point). Therefore, superovulation was used to synchronize, making sure that females were receptive to mating, thereby providing enough cell numbers for the experiment.

(4) Ephx2+ cluster (only present at SO 0.5 dpc and SO estrus) was very interesting. Could the author provide more information about this gene and the potential cell type this cluster represents?

We appreciate the reviewer’s interest in this cell-type cluster. We have now included the discussion regarding this gene, which reads, “Interestingly, the *Ephx2*+ cluster was mainly present in the SO 0.5 dpc and SO estrus samples. *Ephx2* encodes epoxide hydrolase 2, which converts epoxides to dihydrodiols. Recent findings suggest that EPHX2 may play a role in primary hypertension in humans (52). However, the reproductive-related functions of EPHX2 have not yet been investigated. Therefore, we believe this presents an opportunity for future research to define the role of *Ephx2* in the oviduct in response to SO during preimplantation embryo development.” However, as it is beyond the scope of the research provided in this manuscript, we did not further investigate the roles of *Ephx2* in our current study.

(5) "we elucidated whether exogenous hormone treatment impacts protein secretion in the oviduct. There were 298, 354, and 163 differentially abundant proteins when compared between SO estrus vs. SO 269 0.5 dpc". Which hormone?? FSH/LH? Or high estrogens due to more mature follicles; or more embryos instead of hormones? Again, the rationale for using the superovulation model need to be better explained with the consideration of other possibilities.

Thank you for pointing this out. We have clarified that “exogenous hormone treatment” was the superovulation (SO), which is now corrected in the statement, which reads, “we elucidated whether SO treatment impacts protein secretion in the oviduct”.

The justification for the superovulation has now been included in the revised manuscript as indicated in the responses to reviewers above. A detailed description of gonadotropin treatment was included in the Material and Methods section. As the reviewer suggested, we have revised in the Discussion, including the caveat and possibility of the other factors that could lead to biological changes we observed subsequent to SO, which reads,

“As SO increases the number of mature follicles (therefore, estrogen levels), ovulated eggs, and follicular fluid, it is also likely that these biological alterations could lead to changes in the protein abundance in the oviduct”.

(6) "we used RNAScope in situ hybridization staining of Tlr2, Ly6g (leukocytes), and Ptprc (common immune cell marker)." Please indicate what cell types Tlr2 marker was for.

We have now corrected the statement to include the cell types with *Tlr2*+ staining, which reads, “we used RNAScope in situ hybridization staining of *Tlr2* (epithelium, stroma, and myosalpinx)*,* ”.

(7) In which cell types are P38 and p-P38 expressed?

Based on our scRNA-seq searchable dataset, which has been included in the revised manuscript (https://genesearch.org/winuthayanon/Oviduct_pregnancy/), we found that *Mapk14* (encoding P38) was highly expressed in the immune cells in mice (red arrows in the UMAPs below).

In humans, scRNA-seq data published by Ulrich *et al.* (PMID: 35320732) showed that *MAPK14* was present in most cell types in the Fallopian tubes at low levels (see violin plot below).

**Author response image 2. sa3fig2:** 

(8) "Our findings showed an influx of Ptrprc+ cells to the stromal layer, and subsequently penetration into the epithelial layer in the presence of sperm at 0.5 dpc in the UTJ." The authors didn't have results for tracking the influx Ptrprc+ cell to the stromal layer.

Thank you for pointing this out. We agreed with the reviewer’s assessment, as we did not have the results of the tracking of the influx of *Ptprc+* cells. We have corrected and removed the “influx” statement, which now reads, “Our findings showed that *Ptrprc*+ cells were present in the stromal and epithelial layers in the presence of sperm at 0.5 dpc in the UTJ.”

**Reviewer #2 (Public review):**
The manuscript investigates oviductal responses to the presence of gametes and embryos using a multi-omics and machine learning-based approach. By applying RNA sequencing (RNA-seq), single-cell RNA sequencing (sc-RNA-seq), and proteomics, the authors identified distinct molecular signatures in different regions of the oviduct, proximal versus distal. The study revealed that sperm presence triggers an inflammatory response in the proximal oviduct, while embryo presence activates metabolic genes essential for providing nutrients to the developing embryos. Overall, this study offers valuable insights and is likely to be of great interest to reproductive biologists and researchers in the field of oviduct biology. However, further investigation into the impact of sperm on the immune cell population in the oviduct is necessary to strengthen the overall findings.

We appreciate the concise summary, strengths, and weaknesses highlighted. We have addressed all comments made by the reviewer concerning superovulation, figure recommendations, and additional analysis in our revised manuscript. We have included a new analysis of scRNA-seq datasets from human Fallopian tube tissues collected from hydrosalpinx patients and healthy subjects by Ulrich *et al.* (PMID: 35320732). The evaluation of this human data helped distinguish between different inflammatory pathways stimulated by sperm vs. general inflammation, as well as species differences (more details in responses below). In future studies, we will follow up on a detailed description of immune cell types present at 0.5 dpc using FACS analysis. This is mainly due to a lack of expertise and technical limitations in our lab on immune cell investigation. Nevertheless, we have already recruited two immunologists to facilitate our future immune cell studies. We have also provided a clear justification for superovulation, especially in the scRNA-seq analysis in the revised manuscript (please see response to Reviewer 1 above).

**Recommendations for the authors:**
(1) In Figure 3A and 3B, the authors should provide higher contrast and high-resolution images for the expression of the selected immune cell markers at 0.5 dpc and 0.5 dpp. For better clarity and flow, 0.5 dpc & 0.5 dpp, as well as 1.5 dpc & 1.5 dpp, should be merged into a single panel.

Thank you for this suggestion. As shown in the response to Reviewer 1 above, we have now used a higher-magnification image for Fig. 3. We have also changed the panel in the quantification graphs to better reflect the immunofluorescent images and improve clarity and flow.

(2) The authors demonstrated that sperm induces an inflammatory response in the oviduct by presenting IF for selected immune cells. However, FACS analysis should be included to dissect the various immune cell populations further.

We appreciate the recommendation and agree that FACS analysis should provide a more detailed description of the immune cell types present at 0.5 dpc. However, our current work primarily offers initial investigations, confirming that three bioinformatic models (bulk RNA-seq, scRNA-seq, and proteomic analyses) can be validated by IF staining. Our future research using analysis should provide additional characterization of immune cell types at 0.5 dpc in the oviduct.

(3) In Figure 2, the authors performed proteomic analysis at different stages of implantation. They observed similar alterations in the pro-inflammatory Reactome, as seen with RNA-seq and sc-RNA-seq analyses. It would be interesting to examine the types of proteins induced by embryo presence and how their expression changes at 1.5 and 2.5 dpc. Similarly, are sperm-interacting proteins induced in response to sperm presence at 0.5 dpc? Are these proteins uniquely present in the isthmus compared to the ampulla?

We sincerely appreciate the reviewer’s insightful comments regarding the findings in Figure 2 and the potential avenues for further exploration of the proteomic analysis during different stages of embryo preimplantation. We found that during 1.5 dpc, enriched Reactome included Innate Immune System and RHO GTPase (Fig. S4A). In comparison, Reactome at 2.5 dpc were enriched for Keratinization, Metabolism of Protein, and Post-translational Protein Modification (Fig. S4B). Therefore, the pro-inflammatory Reactome profile appeared to have completely subsided at 2.5 dpc. This statement has now been included in the results section, which reads, “Lastly, differential protein abundance at 1.5 dpc and 2.5 dpc indicated the enrichment for Ras Homolog (RHO) GTPase signaling pathway and changes in epithelial remodeling (keratinization) (Fig. S4A and B), respectively. Therefore, the pro-inflammatory Reactome profile appeared to have completely subsided at 2.5 dpc”.

And yes, we detected sperm-interacting proteins (such as OVGP1, ANXA1, HSPA5, and PDIA6, etc.) from our 0.5 dpc proteomic datasets (see examples from images below taken from our dataset: https://genes.winuthayanon.com/winuthayanon/oviduct_proteins/). We noticed that all of these sperminteracting protein levels were lower at 0.5 dpc compared to other timepoints. We speculated that these proteins bind to the sperm and were washed out together with the sperm during the pre-processing centrifugation prior to mass spectrometry analysis. However, we could not distinguish the original location (ampulla vs. isthmus) of proteins as the luminal fluid was flushed from the entire oviduct.

**Author response image 3. sa3fig3:** 

(4) Given that salpingitis is associated with inflammation of the fallopian tubes, the authors should consider comparing the gene signatures from this study with publicly available salpingitis datasets.

Thank you for this insightful suggestion. We have reanalyzed the human data from scRNA-seq of Fallopian tube tissues collected from hydrosalpinx (inflamed and dilated tube) and healthy patients by Ulrich *et al.* (PMID: 35320732). From this published human dataset, we have evaluated GO biological pathways enriched in the differentially expressed genes (DEGs) in hydrosalpinx compared to healthy Fallopian tubes. We have added these new data in the revised Results, Fig. 5 and Supplementary Dataset S5. The new data now read,

“Evaluation of human hydrosalpinx Fallopian tubes compared to sperm-induced inflammation genes

To determine whether sperm-induced inflammatory responses in the mouse oviduct are similar to or different from human inflammation conditions, we reanalyzed publicly available scRNA-seq data from hydrosalpinx samples by Ulrich et al (50)*.* We found that some of the sperm-induced inflammatory genes identified from our mouse study were present and upregulated in hydrosalpinx samples compared to healthy subjects (Fig. 5A). However, the differentially expressed levels, for example the *CCL2* gene, appeared to be marginal between healthy vs. hydrosalpinx samples (Fig. 5_B-C_ and Supplemental Datasets S5). Nevertheless, the top five most enriched GOBPs related to inflammatory responses were Regulation of Complement Activation, Positive Regulation of Macrophage Migration Inhibitory Factor Signaling Pathway, MHC Class II Protein Complex Assembly, Positive Regulation of NK Cell Chemotaxis, and Negative Regulation of Metallopeptidase Activity (Fig. 5D). These GOBPs differed from those identified in sperm-exposed mouse oviducts at 0.5 dpc, which were enriched for neutrophil-related pathways, unlike macrophages or NK cells in hydrosalpinx samples”.

We have also added a revised Discussion, which now reads,

“Lastly, we found that sperm-induced inflammatory conditions in the oviduct were potentially different than those of chronic inflammatory conditions in human Fallopian tubes. The inflammatory responses observed in mice and humans exhibited significant distinction based on immune cell involvement, mechanisms, and context. In mice, acute inflammation after sperm exposure could be primarily characterized by the activation of neutrophils, which serve as the first responders to injury or foreign bodies. In contrast, human Fallopian tubes with hydrosalpinx conditions displayed chronic inflammatory conditions predominantly involving macrophages and NK cells, suggesting a more complex and sustained immune response. It is also possible that inflammation in the oviduct differs between mice and humans. Understanding these species-specific variations is crucial for developing effective therapeutic strategies, as findings from murine models may not accurately translate to human inflammatory conditions due to the distinct immune dynamics at play”.

(5) In Line 259, the authors should clarify why SO females were chosen for luminal fluid collection at different points.

Thank you for pointing this out. We wanted to clarify that the proteomic analysis from the luminal fluid was performed in both naturally mated with and without SO. We have revised the statement in the Results section, which now reads,

“To validate our transcriptomics data at a translational level, LC-MS/MS proteomic analysis was performed on secreted proteins in the oviductal luminal fluid at estrus, 0.5, 1.5, and 2.5 dpc with or without SO. As we also aim to address whether changes in proteomic profiles in the oviduct are governed by hormonal fluctuations, the SO was performed using exogenous gonadotropins. Therefore, the comparison was assessed in the following groups: estrus, 0.5 dpc, 1.5 dpc, 2.5 dpc, SO estrus, SO 0.5 dpc, SO 1.5 dpc, and SO 2.5 dpc”.

In addition, we have now provided additional clarification in the Method section, which reads,

“In this context, our SO approach facilitates multi-dimensional analysis comparisons among naturally cycling bulk RNA-seq, SO scRNA-seq, and natural luminal proteomic biological replicates, enhancing confidence between different methods. This experimental design also reflects adaptive responses in the oviduct during natural fertilization and preimplantation development, influenced by PMSG and hCG treatments at both RNA and protein levels. Furthermore, SO is commonly used in female reproduction to synchronize estrus cycles in animals, thus reducing variables at each collection timepoint.”.

(6) The authors should include scale bars in all fluorescent images.

We apologize for this oversight. In all applicable figures, we have provided a scale bar for all immunofluorescent images.